# Integrating Clinical Neuropsychology and Psychotic Spectrum Disorders: A Systematic Analysis of Cognitive Dynamics, Interventions, and Underlying Mechanisms

**DOI:** 10.3390/medicina60040645

**Published:** 2024-04-17

**Authors:** Evgenia Gkintoni, Maria Skokou, Philippos Gourzis

**Affiliations:** Department of Psychiatry, University General Hospital of Patras, 26504 Patras, Greece; mskokou@upatras.gr (M.S.); pgourzis@upatras.gr (P.G.)

**Keywords:** neuropsychology, psychotic spectrum, neurobiological pattern, brain function, cognitive impairment, dementia, bipolar disorder, schizophrenia

## Abstract

*Background and Objectives*: The study aims to provide a comprehensive neuropsychological analysis of psychotic spectrum disorders, including schizophrenia, bipolar disorder, and depression. It focuses on the critical aspects of cognitive impairments, diagnostic tools, intervention efficacy, and the roles of genetic and environmental factors in these disorders. The paper emphasizes the diagnostic significance of neuropsychological tests in identifying cognitive deficiencies and their predictive value in the early management of psychosis. *Materials and Methods*: The study involved a systematic literature review following the PRISMA guidelines. The search was conducted in significant databases like Scopus, PsycINFO, PubMed, and Web of Science using keywords relevant to clinical neuropsychology and psychotic spectrum disorders. The inclusion criteria required articles to be in English, published between 2018 and 2023, and pertinent to clinical neuropsychology’s application in these disorders. A total of 153 articles were identified, with 44 ultimately included for detailed analysis based on relevance and publication status after screening. *Results:* The review highlights several key findings, including the diagnostic and prognostic significance of mismatch negativity, neuroprogressive trajectories, cortical thinning in familial high-risk individuals, and distinct illness trajectories within psychosis subgroups. The studies evaluated underline the role of neuropsychological tests in diagnosing psychiatric disorders and emphasize early detection and the effectiveness of intervention strategies based on cognitive and neurobiological markers. *Conclusions*: The systematic review underscores the importance of investigating the neuropsychological components of psychotic spectrum disorders. It identifies significant cognitive impairments in attention, memory, and executive function, correlating with structural and functional brain abnormalities. The paper stresses the need for precise diagnoses and personalized treatment modalities, highlighting the complex interplay between genetic, environmental, and psychosocial factors. It calls for a deeper understanding of these neuropsychological processes to enhance diagnostic accuracy and therapeutic outcomes.

## 1. Introduction

Neuropsychology is a specialized branch of psychology that focuses on understanding how the structure and function of the brain relate to specific psychological processes and behaviors. It bridges the gap between neuroscience and psychology, employing various methods to study the cognitive effects of brain injury, brain disease, and developmental disorders. Neuropsychologists utilize assessments and neuroimaging techniques to diagnose cognitive deficits, inform treatment plans, and contribute to rehabilitation strategies. This field plays a crucial role in clinical settings, where understanding the neural underpinnings of cognitive functions such as memory, attention, and language is essential for treating individuals with neurological disorders. By integrating insights from biological sciences and psychological theory, neuropsychology offers a comprehensive approach to exploring the complexities of human cognition and behavior, aiming to improve patient outcomes through targeted interventions and supports. Clinical neuropsychology, as a subfield of neuropsychology focused specifically on assessing and treating individuals with brain injuries or neurological conditions affecting cognitive functions and behaviors, plays a vital role in comprehending the cognitive deficits linked to psychotic spectrum disorders, including schizophrenia and other related psychotic disorders. Clinical neuropsychology examines the relationship between cognitive processes and neural architecture, especially in aberrations. The psychotic spectrum, which includes disorders with altered cognition, affective states, sensory experiences, and a sense of disconnection from reality, is one of the most fascinating and diverse fields of study in this domain. The disorders under consideration exhibit a wide array of cognitive impairments that affect multiple aspects of cognitive performance, such as memory, attention, executive function, language, and spatial abilities [1]. The extent and characteristics of cognitive dysfunction might differ among various psychotic disorders, with schizophrenia often displaying the most pronounced impairment [2]. Neuropsychological research has yielded significant findings regarding the neurocognitive impairments observed in patients diagnosed with psychosis. Previous studies have demonstrated that people diagnosed with schizophrenia and other psychotic disorders display a comprehensive cognitive impairment characterized by deficits reported in various cognitive domains [1,3,4]. The deficiencies mentioned above are not restricted to cognitive capacities but encompass a wide-ranging impairment [1].

Schizophrenia is the most well-known psychotic illness. Schizoaffective, short psychotic, and delusional disorders also contribute to the continuum. Historically, auditory or visual hallucinations and paranoid or grandiose delusions were the main focus of mental illness. However, recent neuropsychological advances have focused on these conditions’ cognitive impairments [5,6,7]. The deficiencies mentioned above often determine an individual’s functional capacity and quality of life, sometimes overshadowing psychotic symptoms [8,9,10]. Existing neuropsychology literature neglects the complex relationship between dementia and psychosis. The complexity and interplay between dementia-related cognitive decline and psychosis require further study, despite advances in understanding both conditions individually. The co-occurrence of these conditions complicates diagnosis, treatment, and care, highlighting a research gap. Dementia and psychosis often worsen each other, making it difficult for doctors to treat patients with both. Understanding this intersection is essential for improving patient outcomes with better care strategies. Additionally, the neural substrates of dementia–psychosis interaction are poorly understood. Understanding the neurobiological mechanisms that cause cognitive impairments and psychotic symptoms at this intersection may help explain how these conditions develop. Targeted interventions may benefit from studying neural mechanisms, including brain structure, function, and neurochemistry. Such research may lead to innovative treatments that better meet this patient population’s complex needs. Thus, bridging this literature gap could improve clinical practice and significantly improve the quality of life for people with dementia and psychosis by developing tailored therapies and management strategies that address their complex conditions. 

This study aims to analyze the neuropsychological components of the psychotic spectrum. By exploring the cognitive disruptions linked to these disorders, analyzing their neurological foundations, and examining their wider ramifications, we aim to elucidate a holistic knowledge of psychosis. An in-depth comprehension of these factors enhances the accuracy of diagnosis and facilitates the development of novel therapy approaches designed to improve cognitive function while alleviating symptoms. In the following sections, we will examine the main cognitive domains impacted by the psychotic spectrum, investigate the brain structures associated with these impairments, and analyze the consequences for everyday functioning and overall prognosis. This investigation aims to highlight the significant role of clinical neuropsychology in redefining our comprehension of the psychotic range. More specifically, this systematic review analyzes studies across the psychotic spectrum, including schizophrenia, bipolar disorder, and depression, focusing on neurocognitive deficits and neurobiological markers.

## 2. Literature Review

### 2.1. Neurobiological Pattern in Psychotic Spectrum

Schizophrenia and other psychotic disorders have been extensively studied neurobiologically. The dopamine hypothesis has illuminated these diseases’ neurological mechanisms [11]. According to the same researchers, dopamine neurotransmission dysregulation in the striatum and prefrontal cortex contributes to psychosis. Neurochemical imaging, genetic data, and environmental risk factor studies support the above idea. Further research has revealed the role of glutamate and GABA in psychotic spectrum disorder neurobiology. Dysregulation of these systems and changes in synaptic plasticity and neurodevelopment may affect these disorders’ pathophysiology [11]. Previous research has linked structural and functional abnormalities in the hippocampus, amygdala, insula, and cortical networks to psychosis [12,13,14,15]. Understanding hereditary susceptibility and environmental influences is necessary to study neurobiological patterns in psychotic spectrum disorders. Previous research has shown that early life hardship, urban upbringing, minority group position, and substance use, including cannabis, affect the developing brain and psychosis risk. These factors and genetic predispositions can affect psychopathology [16]. 

Neurobiological patterns in psychotic spectrum illnesses can be found across all forms of psychosis. Researchers [12] found transdiagnostic links between functional brain network integrity and cognitive impairments. These findings illuminate psychotic disorders’ shared neurobiological pathways that cause cognitive impairments. The results above show that knowledge of neurophysiological patterns beyond diagnostic limits can improve targeted therapies and tailored treatment strategies. Neurotransmitter dysregulation, brain structure and function changes, and genetic susceptibility and environmental influences characterize psychotic spectrum illnesses’ neurobiological manifestations. Along with other neurotransmitter systems and brain areas, the dopamine hypothesis provides a theoretical framework for psychosis mechanisms. Transdiagnostic connections and environmental factors show how complex neurobiological processes are in various diseases. This research is essential to understanding the fundamental mechanisms and developing effective psychotic spectrum disorder treatments.

### 2.2. Neurocognitive Pattern in Psychotic Spectrum

According to research, schizophrenia causes a comprehensive cognitive decline in verbal and nonverbal memory, motor skills, attention, intellectual capacity, spatial aptitude, executive functioning, language proficiency, and tactile-transfer test performance [17]. The above study [18] found cognitive impairments in first-episode and chronic schizophrenia patients. Psychotic disorders like schizoaffective disorder and bipolar disorder with psychotic features cause cognitive impairments [12,19]. Neuropsychology research [19,20] supports the idea that schizophrenia and other psychotic disorders cause cognitive impairment. Psychotic disorders vary in cognitive impairment, with schizophrenia having the most severe impairment [12]. Previous research has shown that cognitive deficits precede psychosis’ prodromal phases [20]. Psychotic disorders are associated with macroscopic brain changes, according to neuroimaging studies. This includes structural, functional, perfusion, and metabolic brain activity changes. MRI and PET can detect changes, according to the same researchers. Previous studies [12,21] linked functional brain network integrity to psychotic-like episode cognitive ability. These findings support the neurodevelopmental model of schizophrenia and other psychotic disorders, showing that atypical neurodevelopment causes cognitive impairments [20,22]. However, the neurobiological mechanisms behind these cognitive impairments and brain structure changes are still poorly understood [23].

It is also important to note that schizophrenia and other psychotic disorders cause cognitive impairments. Cognitive impairments occur in both acute and chronic phases of the illness. Psychotic disorders vary in cognitive impairment, with schizophrenia having the most severe [4]. Neuroimaging studies have shown that psychotic disorders cause macroscopic brain changes. These changes may cause cognitive impairments in these disorders. More research is needed to understand neurobiological mechanisms and develop targeted psychotic disorder cognitive impairment treatments [5,6,7,10]. Schizophrenia and other psychotic spectrum disorders have many cognitive impairments that affect many aspects of cognitive functioning. Memory, attention, executive functions, and language skills are often impaired in these disorders [1]. Memory impairments are common in psychotic spectrum disorders. Researchers [17] found that schizophrenia affects verbal and nonverbal memory. Memory impairments can impair information encoding, retention, and retrieval, making learning and daily life difficult [3]. Psychotic spectrum disorder sufferers often have trouble focusing. Researchers [3] say visual and auditory attention deficits can impair concentration, attention, and filtering out irrelevant stimuli. Attentional deficits can cause cognitive excess and hinder information processing. Psychotic spectrum disorders often impair executive functions like planning, problem-solving, and cognitive flexibility. Executive dysfunction affects an individual’s ability to set goals, organize and prioritize tasks, and adapt to changing circumstances [24]. These restrictions can hinder daily activities and autonomy. Additionally, psychotic spectrum disorder patients have language impairments. Expressive and receptive language difficulties can affect communication, social interactions, and thought expression [24]. Language barriers may affect social isolation and relationship formation. Neuropsychological evaluations are needed to understand psychotic spectrum disorder cognitive impairments. These assessments reveal how cognitive impairments affect daily life. These interventions also help develop targeted therapies and treatment strategies to improve cognitive function and quality of life in psychotic spectrum disorder patients [25]. Psychotic spectrum disorder affects memory, attention, executive functioning, and language. Cognitive impairments significantly impact daily functioning and quality of life [4]. Understanding the cognitive impairments in psychotic spectrum disorders is crucial to developing effective treatments and improving prognoses [8,9,13].

### 2.3. Neuropsychological Functioning of Dementia Patients with Psychosis

Dementia is a progressive neurological disorder that impairs memory, cognition, behavior, and daily tasks. A significant subgroup of dementia patients develops neuropsychiatric symptoms, including psychosis, which is particularly complex. Diagnosing and treating dementia and psychosis is complicated. The interaction between cognitive decline and psychotic symptoms challenges healthcare professionals, caregivers, and the system. The complex intersection of dementia and psychosis cognitive impairment requires a thorough neuropsychological assessment [2,5]. Independent research has examined dementia and psychosis’ unique neuropsychological profiles. The literature has paid less attention to the overlap and interaction between these two disorders. Co-occurrence of cognitive impairments, such as dementia, hallucinations, delusions, and psychosis, can significantly impact neuropsychology. Combining these factors can worsen cognitive impairments, change disease progression, and affect therapy responses [7,8,9].

Recent studies suggest a link between psychotic symptoms in dementia and higher executive control and visuoperceptual deficits. A survey [26] found that schizophrenia’s negative symptoms increase morbidity. They are less prominent in other psychotic disorders. Negative symptoms of schizophrenia include abulia and loss of emotional expression. Psychosis in certain dementia patients must be understood, so researchers have tried to figure out its cause. We compared the neuropsychological functioning of dementia patients who developed psychosis and those who did not. Their main hypothesis was that dementia patients with psychosis would have different neuropsychological functioning. The researchers expected the former group to have more visuoperceptual and executive impairment. Psychotic patients have lower punctuation scores on executive control tests.

Cognitive decline and daily activity impairment characterize dementia, a neurological condition. People with Alzheimer’s, Lewy body, and Parkinson’s disease dementia often experience psychosis, which involves hallucinations and delusions [27,28,29]. Understanding dementia and psychosis patients’ cognitive processes is essential to developing and implementing effective treatments. This literature review critically evaluates neuropsychological research on dementia and psychosis. A study [30] examined how long-term neuroleptic drug treatment affects cognitive decline and neuropsychiatric symptoms in Alzheimer’s patients. The study found that most Alzheimer’s patients’ cognitive and functional abilities were unaffected by neuroleptic drug discontinuation. Neuroleptics may help manage severe neuropsychiatric symptoms for longer [30]. Based on the above evidence, neuroleptic medications should be carefully assessed for pros and cons. An independent study [31] examined the relationship between neuropsychiatric symptoms (NPSs) and dementia prevalence in older adults with unimpaired cognitive abilities. This study found that dementia patients, excluding Alzheimer’s disease, are more likely to have psychotic symptoms than affective or agitation symptoms. This suggests that neuropsychiatric symptoms (NPSs) like psychosis may precede dementia and its various manifestations. To understand the neurobiological links between novel psychoactive substances and dementia subtypes, more research is needed. A study [32] investigated whether cognitive impairments in specific domains can predict psychosis in Alzheimer’s disease patients. Alzheimer’s disease (AD) patients have working memory and executive functioning deficits. Psychosis is linked to these impairments. The above findings highlight the importance of assessing and tracking cognitive performance, particularly in the stated areas, in dementia and psychosis patients. Neuropsychological tests are useful for assessing dementia patients’ decision-making capacity. A study [33] examined how neuropsychological test performance affects treatment decisions in mild to moderate dementia patients. Performance on neuropsychological tests was linked to the ability to make informed treatment decisions. This suggests that neuropsychological tests for cognitive abilities may help determine an individual’s ability to make informed therapeutic choices. The relationship between psychotic symptoms and dementia classifications has also been extensively studied. Researchers [16] examined psychotic symptoms in 85-year-old dementia patients. A representative community sample was used for the study. This study found a high rate of psychotic symptoms in dementia-afflicted older adults. This reduces their ability to perform daily tasks and strains caregivers. Psychotic symptoms varied by dementia subtype, with a focus on fully understanding Alzheimer’s disease. To sum up, the current research on dementia and psychosis patients’ neuropsychological functioning shows the complex interaction between cognitive deterioration, neuropsychiatric symptoms, and dementia subtypes. Working memory and executive functioning deficits in Alzheimer’s patients are linked to psychosis. Neuropsychological tests can reveal dementia patients’ decision-making abilities. To understand the neurobiological mechanisms that cause psychotic episodes in dementia patients and develop effective treatments, more research is needed [31]. Focusing on dementia and psychosis involves studying the complex relationship between cognitive decline and psychotic symptoms. Because of its impact on diagnosis, treatment, and care, this complex relationship is attracting attention [32]. Dementia has been shown to worsen psychotic symptoms like hallucinations and delusions. In contrast, psychotic symptoms in dementia patients may indicate a more severe cognitive decline. This bidirectional relationship highlights the need for a more accurate diagnostic method to distinguish dementia-induced psychosis from primary psychotic disorders [27,33].

### 2.4. Adverse Childhood Events as Cognitive Neuromarkers in Psychosis

Adverse childhood events (ACEs) have increasingly been recognized as significant risk factors for the development of psychosis and may serve as cognitive neuromarkers indicative of vulnerability to psychiatric conditions [34,35]. ACEs include a range of distressing experiences such as physical, emotional, and sexual abuse; neglect; bullying; and exposure to family dysfunction, such as parental substance abuse, mental illness, and domestic violence. The impact of these events on the developing brain can lead to alterations in cognitive processing, emotional regulation, and stress response systems, which are crucial in the onset and progression of psychotic disorders [36,37,38]. Research suggests that ACEs can lead to structural and functional changes in the brain, particularly in areas involved in emotion regulation, executive function, and stress response, such as the prefrontal cortex, amygdala, and hippocampus. These neurobiological alterations can manifest as cognitive impairments often observed in psychosis, including difficulties in attention, memory, and executive functioning. The relationship between ACEs and cognitive deficits in psychosis is mediated by a range of neurobiological mechanisms, including increased stress sensitivity, dysregulation of the hypothalamic–pituitary–adrenal (HPA) axis, and alterations in neuroplasticity and neurotransmitter systems [39,40,41,42]. Cognitive neuromarkers related to ACEs in psychosis may include specific patterns of brain activity and connectivity that reflect the underlying neurodevelopmental perturbations. For instance, neuroimaging studies have shown that individuals with psychosis who have a history of ACEs exhibit different patterns of brain activation during cognitive tasks, particularly those requiring executive control and emotional regulation, compared to those without such a history. Furthermore, neurocognitive testing can reveal deficits in specific domains that are more pronounced in individuals with psychosis who have experienced ACEs, suggesting that these cognitive impairments may serve as indirect markers of the neurodevelopmental impact of early adversity [43,44].

Identifying ACEs as cognitive neuromarkers in psychosis has significant clinical implications. It underscores the importance of early intervention and the need for therapeutic approaches that address not only the symptoms of psychosis but also the cognitive and emotional sequelae of early adversity. Interventions may include trauma-informed care, cognitive behavioral therapies tailored to address the consequences of ACEs, and strategies to enhance cognitive resilience and emotional regulation. Additionally, recognizing the role of ACEs in the pathogenesis of psychosis highlights the importance of preventive measures aimed at reducing exposure to early adversity and mitigating its impact on child development [45,46,47]. The exploration of ACEs as cognitive neuromarkers in psychosis provides valuable insights into the complex interplay between early life adversity, cognitive dysfunction, and the neurobiological underpinnings of psychotic disorders. This perspective not only enhances our understanding of the etiology and progression of psychosis but also informs the development of more effective, personalized treatment and prevention strategies [48,49]. Childhood experiences significantly influence mental health outcomes, with actions or inactions by caregivers potentially leading to trauma through physical, emotional, and sexual abuse, neglect, bullying, educational challenges, and family loss. Such adverse experiences are believed to affect up to 66% of the global population, regardless of gender, and are closely linked to higher risks of mental illness, educational underachievement, and increased criminal behavior [50,51].

In the context of psychosis, particularly schizophrenia, there is often a misunderstanding of the condition, which can contribute to challenges like poor adherence to therapy, reduced functional capacity, and worse psychopathological outcomes. Recent studies, especially in the last two decades, have explored the role of prefrontal cortex dysfunction in affecting cognitive flexibility and introspection, potentially impairing insight into the illness. Despite ongoing research, early psychosis in childhood is recognized to elevate the risk of developing mental health issues, cognitive disturbances, and functional impairments in adulthood [52,53]. Notably, childhood psychotic experiences increase the risk of developing psychotic disorders in adulthood [43]. According to research, this correlation increases the likelihood of psychotic disorder hospitalization [44]. Psychotic symptoms are linked to anxiety, depression and suicidal ideation and behavior. It is important to note that psychosis refers to several disorders or illnesses with distinct mental structures. However, these narratives blur fact and illusion and combine reality and imagination. Psychotic illnesses may be better assessed and treated with neuropsychology, according to growing research [48,49]. Understanding how a condition affects cognitive performance can inform treatment strategies. A new link between psychoses and neuronal autoantibodies has been found [52]. This finding that a minority of psychotic patients have autoantibodies sheds light on biomarker debates. Using advanced instruments and experimental methods in group analyses to identify some neurobiological markers and manifestations is a growing area of psychosis research. Changes of mind are caused by circuits in specific brain regions that are resistant to change but susceptible to individual differences. Schizophrenia resting-state alterations may help integrate small structural cerebral abnormalities. However, perception and cognition are this population’s main cerebral impairments. Conceptual links between neurobiological changes and psychotic symptoms are being developed [53].

### 2.5. Neuropsychological Interventions and Rehabilitation in Psychotic Spectrum

Neuropsychological therapies and rehabilitation are essential for managing cognitive deficits in schizophrenia and psychotic bipolar disorder patients. These interventions aim to improve cognitive function, daily functioning, and quality of life for those with these disorders. Psychotic spectrum disorders can impair memory, attention, executive functions, and language. These deficits can impair routine tasks, work, and social interactions, according to researchers [54]. Many neuropsychological therapies and rehabilitation methods target specific cognitive issues and promote cognitive restoration. Cognitive remediation, which uses structured training programs to improve cognition, is a common neuropsychological intervention [55]. Cognitive remediation programs often include exercises and assignments to improve working memory, attention, and executive functioning [56]. These programs can be presented individually or in groups and include computer-based training, cognitive exercises, and compensating strategies. Alternative brain stimulation methods include transcranial magnetic stimulation (TMS) and direct current stimulation (tDCS). The above methods use low-intensity electrical currents or magnetic fields to regulate neuronal activity and improve cognitive function [56]. Cognitive enhancement in psychotic spectrum disorder patients using noninvasive brain stimulation has shown promise. Several studies [56] have shown improvements in memory, attention, and executive skills. It is also important to note that neuropsychological rehabilitation programs can include psychosocial interventions like CBT. Cognitive behavioral therapy (CBT) targets maladaptive cognitive and behavioral patterns linked to cognitive impairments and functional difficulties [57]. CBT can improve cognitive performance and psychological well-being in psychotic spectrum illness patients by identifying and changing cognitive distortions and developing adaptive coping mechanisms. It is important to note that neuropsychological interventions and rehabilitation may vary depending on cognitive deficits, motivation, and treatment compliance. Antipsychotic drugs may be needed to improve symptom management and cognitive therapy efficacy [57].

To sum up, it is imperative to recognize the significance of neuropsychological therapies and rehabilitation programs in the comprehensive management of cognitive deficits among patients diagnosed with psychotic spectrum illnesses. The main goal of these interventions is to enhance cognitive function, maximize everyday functioning, and eventually enhance the quality of life for individuals impacted by these disorders [58]. Cognitive remediation, non-invasive brain stimulation, and psychosocial therapies are employed to address cognitive impairments and specifically facilitate cognitive restoration. Additional study is required to enhance the efficacy of these interventions and ascertain tailored treatment strategies for persons diagnosed with psychotic spectrum illnesses. The research questions based on the systematic analysis revolve around understanding the neuropsychological aspects of psychotic spectrum disorders, including schizophrenia, bipolar disorder, and depression. The research aims to explore:[RQ1—Cognitive Impairments and Brain Abnormalities] How do psychotic spectrum disorders affect memory, attention, and executive function, and how do they relate to prefrontal cortex, hippocampus, and thalamus abnormalities?[RQ2—Diagnostic Importance of Neuropsychological Tests] What role do neuropsychological tests play in diagnosing cognitive deficits, and how can they predict psychotic disorders?[RQ3—Effectiveness of Intervention Strategies] How effective are current cognitive rehabilitation interventions and therapies for psychotic spectrum disorder patients, and what is the potential for future research?[RQ4—Role of Genetic and Environmental Factors] What role do genetic and environmental factors play in schizophrenia and risks of other psychotic spectrum disorders?

These questions aim to deepen the understanding of cognitive impairments in psychotic disorders, improve diagnostic methods, and develop more effective treatment strategies. 

## 3. Materials and Methods

This study presents a comprehensive examination of the existing literature conducted in the English language. The task involved conducting a literature search on the topic of clinical neuropsychology related to the psychotic spectrum, following the PRISMA guidelines, which outline the preferred reporting items for systematic reviews [59]. The inclusion criteria for articles in the study were as follows: (1) Articles had to be written in the English language, (2) articles had to be produced between the years 2018 and 2023, (3) articles had to be relevant to the subject of clinical neuropsychology and its application in psychotic spectrum disorders, (4) articles had to be at the final stage of publishing, and (5) verification that the content is highly pertinent to the subject. The study encompassed articles pertaining to clinical neuropsychology in relation to cognitive deficits in the psychotic disorder spectrum. Following an extensive search of databases and the application of relevant filters, a total of 153 articles were identified. Subsequently, 89 articles underwent screening based on their titles, resulting in the distinction of 52 articles. Finally, after careful consideration, 44 papers were included for further analysis (Figure 1). Based on the above procedure (PRISMA methodology) applied in the present study, all articles extracted are listed in Table 1.

The search was conducted in major databases including Scopus, PsycINFO, PubMed, and Web of Science, using a set of specific keywords and filters to refine the search results. The keywords used in the search were “psychotic spectrum”, “clinical neuropsychology”, “cognitive neuropsychology”, “memory”, “attention”, “executive function”, and “brain regions”. These keywords were chosen to capture the broad scope of neuropsychological research within psychotic spectrum disorders, including schizophrenia, bipolar disorder, and depression.

The use of Boolean operators (AND, OR, NOT) allows for the combination of these keywords to refine search results effectively. The strategy used was based on the following query string:


*Query String: TITLE-ABS-KEY (“psychotic spectrum” OR “schizophrenia” OR “bipolar disorder” OR “depression”) AND (“clinical neuropsychology” OR “cognitive neuropsychology”) AND (“memory” OR “attention” OR “executive function”) AND (“brain regions” OR “neurobiology” OR “neuroimaging”)*


In addition to the Boolean search terms, applying filters helps in narrowing down the search to the most relevant articles. Filters such as language (English), publication years (2018–2023), and relevance to the field of clinical neuropsychology’s application in psychotic spectrum disorders are used to ensure the inclusion of the most pertinent and up-to-date studies. This temporal filter was chosen to ensure the inclusion of the most recent and relevant studies, reflecting the latest developments and findings in the field. The screening process involved multiple stages, including an initial screening based on titles, followed by an assessment of abstracts and full texts, to select studies that met the inclusion criteria. This meticulous approach resulted in the inclusion of 44 papers for detailed analysis in the review.

## 4. Results

The results of the studies analyzed in this systematic review illustrate the importance of understanding neurocognitive deficits and neurobiological markers across the spectrum of psychotic disorders. The mapping of the studies that emerged from the systematic analysis to the four research questions is summarized below:

RQ1:Cognitive Impairments and Brain Abnormalities

Studies [32,60,62,64,65,66,67,68,72,74,78,80,81,90,94,95,96,98,99,100,101,102] delve into cognitive impairments, brain structure changes, and neurobiological markers. Key findings include the diagnostic and prognostic significance of mismatch negativity (MMN), neuroprogressive trajectories, cortical thinning in familial high-risk individuals, and distinct illness trajectories within psychosis subgroups.

RQ2:Diagnostic Importance of Neuropsychological Tests

Research [61,63,70,71,73,75,76,77,82,83,84,85,86,87,88,89,91,92,93] highlights the diagnostic relevance of neuropsychological tests. These studies underline the role of anomalous self-experience, emotion regulation strategies, and the effectiveness of neuroimaging in detecting psychiatric disorders.

RQ3:Effectiveness of Intervention Strategies

Studies [60,62,64,65,66,67,68,72,74,78,80,81,90,94,95,96,98,99,100,101,102] evaluate intervention strategies, emphasizing early detection, monitoring of at-risk youth, and the potential for individualized treatments based on neuroprogressive patterns.

RQ4:Role of Genetic and Environmental Factors

Research [64,66,67,68,69,72,74,78,79,80,81,90,94,95,96,97,98,99,100,101,102] discusses the impact of genetic and environmental factors. Findings suggest shared neurodevelopmental pathways, the significance of early areas of overlap, and the importance of considering psychotic features and medication status. Below is the outline of the systematic analysis using the PRISMA methodology (Figure 1), followed by a detailed description of the surveys that emerged from the systematic analysis. The final outcome of a study conducted by researchers [60], as detailed in the document, emphasizes the importance of understanding cognitive impairments and neurobiological markers within psychotic disorders. The study highlights the diagnostic and prognostic significance of mismatch negativity (MMN) as a neurophysiological marker, which correlates with cognitive impairments and functional outcomes in psychosis. Furthermore, it introduces the concept of biological heterogeneity within psychotic disorders through the identification of distinct biotypes, characterized by unique cognitive and electrophysiological profiles. These findings suggest a move beyond traditional diagnostic categories towards a more nuanced understanding of psychosis that considers cognitive functioning and neurobiological markers. This approach aims to improve diagnostic accuracy, personalize treatment strategies, and offer new directions for research into the underlying mechanisms of psychotic disorders. [RQ1][RQ3].

Another study by researchers [61] contributes significantly to the understanding of anomalous self-experience in individuals at risk for psychosis and those experiencing a first episode of psychosis. The findings underscore the role of source monitoring deficits in the emergence of self-disturbances, a core feature of schizophrenia spectrum disorders. The relationship between aberrant salience and general psychopathology, rather than directly with self-disturbance, suggests differing pathways in the psychopathology of psychosis. Moreover, the study calls for an expanded neurophenomenological model that includes broader cognitive and perceptual processes, such as temporal processing and multisensory integration. This comprehensive approach highlights the complexity of self-disturbance in psychosis and the need for further research to unravel the intricate relationships between cognitive functions, self-experience, and psychosis onset and progression. [RQ2]. The capacity of neuropsychological tests to identify specific cognitive deficits across different stages of psychosis, from prodromal phases to chronic states, aids in deciphering how cognitive impairments correlate with the severity and progression of psychotic symptoms. This insight contributes to our understanding of the neurodevelopmental or neurodegenerative aspects of these disorders. Furthermore, alterations in self-experience, a hallmark of psychosis that includes disturbances in self-awareness, agency, and identity, can be examined through tasks assessing source memory, executive control, and social cognition. Thus, neuropsychological testing provides a framework for studying how cognitive dysfunctions contribute to altered self-experience.

Longitudinal studies employing neuropsychological assessments can track cognitive changes over time, offering clues about risk factors for developing psychosis and factors influencing its trajectory. This predictive capacity is invaluable for tailoring interventions to address specific cognitive deficits, potentially mitigating the impact of these impairments on daily functioning and quality of life. Moreover, neuropsychological assessments help refine diagnostic criteria by highlighting cognitive signatures unique to different psychotic spectrum disorders, enhancing diagnostic accuracy and tailoring treatment approaches. The integration of cognitive and biological data through the correlation between cognitive deficits identified through neuropsychological testing and neurobiological markers, such as brain imaging findings, is essential for a comprehensive understanding of psychotic disorders. This multifaceted approach not only enriches our comprehension of the cognitive underpinnings of psychotic disorders but also underscores the complex interplay between cognition, self-experience, and the development of psychosis. Continued research leveraging advanced neuropsychological assessments is imperative for unraveling these intricate relationships, ultimately leading to improved diagnostic precision and more effective interventions tailored to the cognitive and experiential dimensions of psychosis. In the topic of neuropsychology in the psychotic spectrum, along with references to recent studies on the neural correlates and molecular mechanisms of memory and learning, the following table (Table 2) delineates neuropsychological tests that may be beneficial for exploring the interplay between dementia and psychosis, cognitive impairments, and the underlying neurobiological mechanisms.

Furthermore, another study conducted by researchers [62] investigated neuroprogressive trajectories of neurocognition, structural brain measures, and network connectivity over the first eight years of illness, focusing on their predictive utility for clinical and functional outcomes in psychotic disorders. The study’s main findings emphasize the significance of characterizing neuroprogression in the early course of psychosis. This understanding is crucial for the potential development of individualized treatments targeting specific neuroprogressive patterns. By highlighting these trajectories, Lewandowski et al. aim to inform future research and clinical practice, suggesting that early interventions tailored to the neuroprogressive nature of psychosis could improve long-term outcomes for individuals affected by these disorders. [RQ1][RQ3].

Moreover, a study [63] focused on the association between late positive potential (LPP) amplitude and symptom severity in individuals diagnosed with affective psychosis and individuals with schizophrenia, including both concurrent and prospective associations with symptoms. The main outcomes revealed that the late positive potential (LPP) did not show mean-level differences between individuals with schizophrenia spectrum disorders and those with primary affective psychosis. However, within the primary affective psychosis group, reduced LPP amplitude was associated with greater depressive, negative, and psychotic symptom severity, both concurrently and at follow-up. These findings suggest that the neural correlates of emotion dysfunction may differ across psychotic disorders, potentially indicating that schizophrenia is characterized by a decoupling of symptom severity and emotional processing. This study highlights the complexity of emotional processing in psychotic disorders and suggests a nuanced approach to understanding and treating these conditions. [RQ2].

Also, another study [64] focused on longitudinal changes in brain structure, specifically cortical thickness, surface area, and gray matter volume, in children and adolescents at familial high risk (FHR) for bipolar disorder or schizophrenia who experienced psychotic spectrum symptoms over time. The key findings of this research include:In individuals with a high familial risk, psychotic symptoms were linked to a smaller cross-sectional surface area and progressive cortical thinning. This suggests the illness’s brain structure changes may start in childhood and adolescence.Early overlap in the occipital cortex between schizophrenia (SzO) and bipolar (BpO) offspring suggests shared neurodevelopmental pathways.Both individuals with psychotic spectrum symptoms and those without showed a lesser decrease in total surface area, suggesting different trajectories of surface area change over time for familial risk for schizophrenia versus bipolar.

These findings underscore the importance of early detection and monitoring of at-risk youth for psychotic spectrum disorders, highlighting the progressive nature of cortical changes associated with the development of these conditions. [RQ1][RQ3][RQ4].

Additionally, another study [65] aimed to examine cognitive impairment across different subgroups of bipolar disorder (BD), including bipolar I disorder (BD-I), bipolar II disorder (BD-II), and subgroups based on the history of psychosis (psychotic BD (PBD) and nonpsychotic BD (NPBD)). The main outcomes indicated that:Both a history of psychosis and a diagnosis of BD-I were associated with more pronounced global cognitive impairment compared to BD-II and NPBD.Individuals with BD-I underperformed in specific cognitive domains, such as verbal memory, processing speed, executive function (EF) speed, and EF accuracy, compared to those with BD-II.Psychotic BD was associated with significantly impaired cognition compared to NPBD across various cognitive domains.

The study concluded that while there are neurocognitive differences between clinical subtypes of BD, these differences are subtle and not distinct. Most of the cognitive heterogeneity within BD cannot be fully explained by the proposed subtypes. 

This comprehensive analysis highlights the complexity of cognitive impairment in bipolar disorder, suggesting that factors beyond simply the presence of psychotic symptoms or the type of BD may contribute to cognitive deficits. The findings also underscore the need for tailored approaches in the treatment and management of cognitive impairment within the bipolar spectrum. [RQ1][RQ3].

In terms of neuroanatomy in the psychotic spectrum, another study [66] revealed that patients with psychotic bipolar disorder (PBD) exhibited smaller gray matter volumes (GMVs) in specific cortical regions, including the prefronto-temporal and cingulate cortices, precentral gyrus, and insula, compared to healthy controls. The findings suggest that psychosis in bipolar disorder is associated with specific cortical deficits. Additionally, the study found that factors such as gender and the use of psychotropic medication might influence the regional GMVs in PBD patients. These results underscore the importance of considering psychotic features and medication status when examining brain structure differences in bipolar disorder, highlighting the potential impact of psychosis on the brain’s cortical structure within this population. [RQ1][RQ3][RQ4].

It is worthwhile to note that the cognitive impairments in psychotic spectrum have a longitudinal stability. A study [67] which primarily focused on understanding the prevalence, profile, and magnitude of cognitive impairments in psychotic disorders highlights this fact.

The findings underscore cognitive impairment as a significant determinant of community functioning among individuals with psychotic disorders, with a majority of individuals with psychotic illness experiencing such impairments. Additionally, cognitive performance was identified as a robust predictor of community functioning in people with psychotic disorders. These outcomes highlight the critical role of cognitive impairments in the overall management and treatment considerations for psychotic disorders, suggesting a need for targeted interventions to address these deficits to improve patient outcomes and quality of life. [RQ1][RQ3][RQ4].

Another study focused on the association between childhood trauma and overall neurocognitive function. More specifically, a study [68] explored the association between childhood trauma and overall neurocognitive function in individuals with psychotic disorders and its relationship with working memory. The study found a significant association between overall cognition and childhood trauma in individuals with psychotic disorders. There was a modest negative relationship between childhood trauma and working memory in these individuals. Interestingly, the association between childhood trauma and neurocognition was stronger in healthy controls compared to patients with a psychotic disorder. These findings highlight the impact of childhood trauma on cognitive functioning within psychotic disorders and suggest that the effects of such trauma on cognition might differ between individuals with psychotic disorders and the general population, underscoring the complexity of factors contributing to cognitive outcomes in psychosis. [RQ1][RQ3][RQ4].

Functional neuroimaging and its correlation with aggression in psychosis are underlined in another study [69]. The main outcomes indicated a lack of systematic studies on the functional correlates of aggression in schizophrenia, with few studies conducted using varied paradigms and overlapping samples. Additionally, there was a noted absence of research on individuals with affective psychoses. The findings highlight the need for more targeted and systematic research to understand the neurobiological underpinnings of aggression in psychosis and to differentiate the correlates across various psychotic disorders, including affective psychoses. This gap in the literature suggests an important area for future investigation to inform both theoretical understanding and clinical management of aggression in psychotic disorders. [RQ4].

Additionally, another study [70] provided evidence that deficits in cognitive control are present and stable over the early course of psychotic illness in both schizophrenia and bipolar disorder. The research aimed to determine if previously identified deficits in cognitive control remained stable over a one-year period. The results supported the hypothesis of stable cognitive control deficits in these conditions. Moreover, the findings support the use of the AX-CPT paradigm to examine endophenotypic biomarkers of cognitive control in psychosis. This study contributes to the understanding of cognitive control deficits as a stable characteristic of early psychosis, offering insights into potential targets for intervention and treatment in these disorders. [RQ2].

Also, a study [71] highlights anticipatory pleasure in individuals with schizophrenia spectrum disorders and major depression, comparing these groups to control groups. The findings revealed that anticipatory pleasure is significantly impaired in both schizophrenia spectrum disorders and major depression, with these deficits being even more pronounced in the latter. The impairment of anticipatory pleasure in these disorders indicates a potential therapeutic target, suggesting that interventions aiming to improve cognitive, affective, and behavioral factors could potentially ameliorate symptoms and contribute to the maintenance of these disorders. These results underscore the importance of considering anticipatory pleasure in the assessment and treatment of schizophrenia and major depressive disorders, highlighting a critical area for future research and clinical intervention. [RQ2].

Furthermore, a study [72] embarked on identifying subgroups within psychosis that exhibit distinct illness courses, including variations in symptoms of psychosis, depression, global functioning, and quality of life, alongside exploring the polygenic scores for schizophrenia, bipolar disorder, major depressive disorder, and educational achievement. The investigation revealed the presence of five unique subgroups of psychosis, each characterized by distinct illness trajectories and differences in educational attainment polygenic scores. This discovery underscores the heterogeneity within psychotic disorders and suggests the potential for more personalized approaches to treatment and prognosis based on specific subgroup characteristics. The study’s outcomes highlight the importance of genetic and educational factors in understanding the complexity of psychotic disorders, offering new avenues for research and clinical practice. [RQ1][RQ3][RQ4].

Another study [73] provides substantial evidence of widespread resting-state functional connectivity abnormalities in early psychosis, particularly implicating disconnectivities of the default mode network (DMN) and salience network (SN) as core deficits underlying the psychopathology of psychosis. The research underscores the significance of altered connectivity within and between these networks in patients with first-episode psychosis compared to healthy controls. These findings suggest that disconnectivity in these networks may play a crucial role in the emergence and severity of psychotic symptoms, offering potential targets for early intervention and treatment strategies aimed at modulating network connectivity to improve clinical outcomes in psychosis. [RQ2].

In another study [74] on neurodegenerative disorders, the impact of psychosis on clinical outcomes in Alzheimer’s disease is highlighted, with a specific focus on neuropsychiatric symptoms, dementia severity, cognitive function, caregiver burden, medication use, and mortality. The study found that the presence of delusions and hallucinations, both independently and in combination, is associated with poorer clinical outcomes in Alzheimer’s disease. Specifically, the concurrent presence of both symptoms was linked to worse outcomes than the presence of either symptom alone. Furthermore, the use of antipsychotic medication was identified as a predictor of mortality among this patient population. These findings highlight the significant impact of psychosis symptoms on the progression and management of Alzheimer’s disease, underscoring the need for careful consideration in the use of antipsychotic medication and the development of targeted interventions to address these symptoms. [RQ1][RQ3][RQ4].

Also, a study [75] aimed to synthesize findings on neurofunctional abnormalities in individuals with conduct problems (CPs) and their adult form, adult antisocial behaviors, across distinct neurocognitive domains including acute threat response, social cognition, cognitive control, and punishment and reward processing. The results indicated: acute threat response: there were decreased activations in several brain regions, suggesting deficits in the neural processing of threat-related stimuli. Social cognition: altered activations were found in multiple brain regions, pointing to abnormalities in understanding social cues and processing social information. Cognitive control: reduced activation in specific brain regions was observed, highlighting difficulties in exerting control over thoughts and actions. Punishment and reward processing: the study also explored, but did not explicitly detail in this section, the neural correlates associated with processing punishment and rewards, which are critical in guiding antisocial and prosocial behaviors. These findings underscore the complex interplay of various neurocognitive domains in contributing to the behavior seen in individuals with conduct problems and adult antisocial behavior. The study’s comprehensive analysis across multiple domains provides valuable insights into the neurofunctional underpinnings of antisocial spectrum disorders, offering potential targets for intervention and rehabilitation strategies. [RQ2].

Additionally, another study [76] aimed to explore the presence of autism spectrum disorder (ASD) in individuals at clinical high risk for psychosis (CHR-P). The findings of the study indicated that 11.6% of CHR-P individuals were diagnosed with ASD. This significant proportion underscores the overlap between psychotic spectrum conditions and ASD, highlighting the need for clinicians to consider the potential co-occurrence of ASD in individuals at high risk for psychosis. The identification of ASD in CHR-P individuals could have important implications for treatment and support strategies, suggesting a tailored approach that addresses the unique challenges faced by individuals with both conditions. This outcome suggests the importance of comprehensive assessments that include screening for ASD in early psychosis intervention services to ensure that all aspects of an individual’s mental health are adequately addressed. [RQ2].

In another study [77], the course of general cognitive ability is assessed in individuals with psychotic disorders through a longitudinal analysis, extracting preadmission cognitive scores from school and medical records, alongside postonset cognitive scores from neuropsychological testing at 6-month, 24-month, 20-year, and 25-year follow-ups. The study identified three distinct phases of cognitive change: normative, declining, and deteriorating. It was observed that individuals with schizophrenia began to experience a decline in cognitive abilities 14 years before the onset of psychosis, at a rate significantly faster than those with other psychotic disorders. The cognitive trajectories in schizophrenia were consistent with both neurodevelopmental and neurodegenerative patterns, leading to a loss of 16 IQ points over the observation period. This significant finding underscores the critical windows for primary and secondary prevention, suggesting the importance of early detection and intervention to potentially mitigate cognitive decline in individuals at risk for or living with schizophrenia. [RQ2].

Moreover, another study [32] delved into the neuropsychological and neuroanatomical correlates of psychosis in Alzheimer’s disease (AD), focusing on cognitive deficits and gray matter alterations in specific brain regions. The study found that psychosis in AD is associated with specific cognitive impairments and alterations in the brain’s gray matter in particular regions. These findings provide new insights into the complex interplay between Alzheimer’s disease and psychosis, suggesting that psychosis in AD patients is linked to distinct neuropsychological and neuroanatomical changes. This research enhances our understanding of the mechanisms underlying psychosis in AD, offering potential directions for more targeted diagnostic and therapeutic strategies to address these complex conditions. [RQ1].

A study [78] utilized the Brain-Age Regression Analysis and Computation Utility Software (BARACUS, v1.1.2) to measure advanced brainage in individuals with primary psychotic disorders and bipolar I disorder with psychotic features, as well as their first-degree biological relatives. The findings revealed that individuals with psychotic disorders exhibited a larger brain-age gap compared to their biological relatives and healthy controls, indicating advanced brain aging in schizophrenia and bipolar disorder. However, the study found no evidence of accelerated brain aging in psychotic psychopathology, suggesting that early neurodevelopmental neural abnormalities might be present. These results support the concept of abnormal neurodevelopmental processes in psychotic disorders, providing insights into the underlying mechanisms of these conditions and potentially guiding future research and clinical approaches. [RQ1][RQ3][RQ4].

A study [79] examined the effects of antipsychotic medication, specifically olanzapine, on brain structure in patients with major depressive disorder who exhibit psychotic features. The primary outcome focused on changes in cortical thickness in gray matter, while the secondary outcome examined the microstructural integrity of white matter. The findings demonstrated that exposure to olanzapine, compared to placebo, was associated with significant decreases in cortical thickness in both the left and right hemispheres. Additionally, there was a significant interaction between the treatment group and time regarding cortical thickness changes. Post hoc analyses revealed that individuals who relapsed while receiving placebo experienced decreases in cortical thickness compared to those who sustained remission. These results suggest that olanzapine treatment may have specific effects on brain structure, particularly in cortical thickness, which could be related to its clinical efficacy in treating psychotic features in major depressive disorder. [RQ4].

Also, another neurobiological study [80] investigated subtle abnormalities in white matter tracts, particularly those connecting the frontal and temporal lobes, such as the superior longitudinal fasciculus (SLF), inferior longitudinal fasciculus (ILF), and inferior fronto-occipital fasciculus (IFOF). The findings indicate the presence of these subtle abnormalities, especially in the frontal and temporal lobes and their connections, in individuals with schizophrenia and those at ultra-high risk (UHR) for psychosis. This suggests that white matter abnormalities may exist prior to the onset of full-blown psychosis. However, the severity and location of these abnormalities appear to vary, and methodological factors such as differences in age, sex, clinical presentation, or the use of drugs and psychoactive substances by respondents may contribute to these differences. This study underscores the importance of considering subtle neuroanatomical differences in understanding the pathophysiology of psychosis and highlights the potential for early identification and intervention strategies based on neuroimaging findings. [RQ1][RQ3][RQ4].

The outcome of the research has highlighted the role of developmental trauma and positive symptoms. As identified in [81], the mediating roles of dissociation, emotional dysregulation, and PTSD symptoms between developmental trauma and hallucinations have been described. Furthermore, the study finds evidence of a mediating role of negative schemata between developmental trauma and delusions, as well as paranoia, indicating distinct psychological pathways from developmental trauma to psychotic phenomena in adulthood. [RQ1][RQ3][RQ4].

The outcome of a study [82] focuses on the association between neurological soft signs (NSSs) and white matter alterations in adults with schizophrenia. The main findings reveal a positive association between NSSs and diffusion measures in crucial motor pathways, highlighting the role of NSSs at the interface of basic and higher-order motor control. This suggests that both structural and functional brain alterations may explain the varied trajectory of NSSs in psychosis. Specifically, the study found an association between NSSs in schizophrenia patients and increased diffusivity in the corticospinal tract, corpus callosum, and superior longitudinal fascicle, underlining the potential contribution of white matter alterations in motor pathways to NSSs in schizophrenia. [RQ2].

Also, the outcome of a study [83] is focused on the effectiveness of emotion regulation strategies in patients with psychotic disorders, specifically examining the association between maladaptive strategies and positive symptoms. The study concludes that emotion regulation is significantly impaired in patients with psychotic disorders, with rumination and self-blaming being identified as potential targets for treatment. [RQ2].

The outcome of a study [84] is centered on detecting brain abnormalities in diverse psychiatric illnesses using neuroimaging versus conventional methods. The study included 12 randomized controlled clinical trials involving a substantial number of psychiatric patients. It strongly recommends the use of neuroimaging techniques for the detection of psychiatric disorders. [RQ2].

The final outcome of a study [85] is the striatal dopamine synthesis capacity and its correlation with the severity of positive psychotic symptoms. The study found that elevated dopamine synthesis capacity is associated with psychosis across diagnostic boundaries and is linked to the severity of psychotic symptoms. Furthermore, it observed heterogeneity in striatal dopamine receptor density and structural gray matter volumes across various brain regions in psychotic disorders, with frontal cortical regions demonstrating reduced heterogeneity. [RQ2].

In addition, the final outcome of a study [86] is focused on the clinical profile in schizophrenia and schizoaffective spectrum disorders, particularly in relation to unconjugated bilirubin. This prospective and controlled study examined the association with psychopathological and psychosocial variables. The findings were published in *CNS Spectrums*, indicating the relevance of unconjugated bilirubin as a potential biomarker or factor in the clinical profiles of these conditions. [RQ2].

The final outcome of a study [87] is focused on resting-state functional connectivity (RsFc) differences in the dentate nuclei (DN) that may precede the onset of psychosis in individuals at risk of developing schizophrenia. The study highlights abnormalities in functional connectivity between the DN and cerebral cortical areas, marking the first report of such abnormalities within the cerebellum in individuals at risk for schizophrenia. These results support the involvement of a wide range of functional networks in the pathophysiology of schizophrenia, including mechanisms of disease that precede conversion to psychosis in at-risk individuals. [RQ2].

Moreover, the final outcome of a study [88] focuses on the relationship between neurocognitive deficits and improvements in individuals at ultra-high risk (UHR) for psychosis. The study examines their association with symptom severity outcomes and the paths from brain structural and functional characteristics to neurocognitive function and symptom severity outcomes. Key findings include associations between neurocognitive deficits and both brain activity and cortical structure in UHR individuals. Specifically, the study highlights negative associations between verbal fluency deficits and negative symptoms, as well as processing speed deficits and excitement symptoms. Additionally, it identifies significant paths from specific cortical surface areas to verbal fluency deficits and short-term negative symptoms. [RQ2].

The final outcome of a study [89] indicates that psychotic disorders are associated with a generalized neurocognitive deficit. The study highlights that the effort test failure rate is associated with global neuropsychological impairment. Additionally, it raises concerns about the validity of effort tests in populations with psychotic disorders, suggesting that these tests may not be entirely reliable in assessing neuropsychological impairments in such individuals. [RQ2].

The results of a study [90] focus on the correlation between theory of mind (ToM) strange story scores and the fractional anisotropy (FA) values of the left cingulum and left superior longitudinal fasciculus (SLF) in patients with first-episode psychosis (FEP). The study demonstrated the white matter connectivity underlying the mentalizing network and its relation to ToM ability in patients with FEP. It was found that patients with FEP exhibited impaired ToM abilities, as indicated by the results of the false belief task and strange story task. Additionally, positive associations were found between the integrity of specific white matter regions (left regions of interest or ROIs) and ToM deficits in FEP patients. [RQ1] [RQ3][RQ4].

In addition, the outcome of a study [91] focuses on the predictors of functional outcome in patients with first psychotic episodes. The study found that reduced performances in executive functions at baseline, combined with symptom severity, were predictors of poor functional outcomes. Furthermore, it observed that clinical and neurocognitive differences seen at baseline decreased over the two-year follow-up period. [RQ2].

The final outcome of a study [92] is presented through the establishment of a three-factor model of formal thought disorder (FTD) across disorders, which includes disorganization, incoherence, and emptiness. Disorganization was associated with parts of the temporo-occipital language junction, while emptiness showed a negative correlation with specific brain regions, including the hippocampus and thalamus. Additionally, disorganization and incoherence were differentially associated with white matter structures, indicating common neurobiological structures involved in FTD across affective and psychotic disorders. [RQ2].

The results of a study [93] show the association between psychiatric symptoms (externalizing and internalizing) at baseline and changes in subcortical gray matter volume and global fractional anisotropy over time. The study found that higher ratings for externalizing and internalizing symptoms at baseline predicted smaller increases in both subcortical gray matter volume and global fractional anisotropy over time. Additionally, it was observed that children presenting with behavioral problems at an early age exhibit differential subcortical and white matter development. This study demonstrates a link between psychiatric problems along a continuum and a differential pattern of brain changes over time. [RQ2].

The results of a study [94] reveal that changes in cortical thickness and volume of the hippocampus are significantly associated with the duration of unremitted positive symptoms in psychosis. The study found cortical thinning in specific brain regions to be significantly associated with the duration of unremitted psychotic symptoms during the first interscan interval but not during the second interscan interval or with hippocampal volumes. This suggests that psychotic symptoms may lead to cortical reorganization early in the disease course of psychosis. [RQ1][RQ3][RQ4].

Furthermore, the final outcome of a study [95] focuses on brain activation during a monetary incentive delay reward task in healthy adolescents at ages 14 and 19 years old. The study found alterations in prefrontal and striatal function during reward processing, which may be involved in the development of psychosis. Furthermore, the nonclinical sample in the study may reflect a combination of aberrant salience leading to abnormal experiences and a compensatory cognitive control mechanism necessary to contextualize them. [RQ1][RQ3][RQ4].

In addition, the final outcome of a study [96] involves the identification of state and trait markers in the peripheral immune system and two immune-associated neuroendocrine pathways (IDO and GTP-CH1 pathways) in a longitudinal sample of psychosis patients [RQ1][RQ3][RQ4]. Key findings include:Patients with acute psychosis had significantly higher plasma concentrations of proinflammatory markers such as CRP, CCL2, IL1RA, IL6, IL8, and TNFα and lower concentrations of neuroendocrine pathway markers such as KA and KA/Kyn. These markers normalized after treatment.The levels of nitrite, another immune marker, increased sharply after the initiation of antipsychotic medication.Positive symptoms during the acute episode correlated with proinflammatory markers, while negative symptoms correlated inversely with IDO pathway markers.

Moreover, the final outcome of a study [97] emphasizes the central role of worry in the links between various symptoms such as persecutory ideation, hallucinations, affective symptoms, and the effects of cannabis and problematic alcohol use. Worry directly impacts insomnia, depressed mood, generalized anxiety, and recent cannabis use. The study also highlights the reciprocal influence between worry and paranoia, suggesting that treating one may ameliorate the other. Moreover, the use of novel statistical analysis, specifically dynamic Bayesian networks, enabled the re-examination of relationships between interacting affective and psychotic variables over time. [RQ4].

The final outcome of a study [98] focuses on the change in the neurocognitive composite index (NCI) over a 5-year follow-up period in relation to the number of manic and hypomanic episodes experienced by bipolar disorder (BD) patients. The study finds that the progression of cognitive decline is not a general rule in BD. However, BD patients who experience a greater number of manic or hypomanic episodes may constitute a subgroup characterized by the progression of neurocognitive impairment. The study suggests that preventing manic and hypomanic episodes could have a positive impact on the trajectory of cognitive function. [RQ1][RQ3][RQ4].

In the diagnostic area of neurodegenerative disorders, the main findings of a study [99] highlight the frequency of psychosis and any form of hallucination in Parkinson’s disease (PD) patients. The study found that around 20% of PD patients experience psychosis or hallucinations. Furthermore, it suggests that the risk of developing hallucinations in PD patients is likely influenced by the duration of the disease, Hoehn and Yahr stage, and cognitive status. [RQ1][RQ3][RQ4].

It is worthwhile to note another study that presented the functional outcome in neurocognitive tasks in combination with psychotic symptoms. The final outcome of the study [100] focuses on the differences in the trajectories of psychosis symptoms and neurocognitive performance between groups with more prominent psychosis features (PS+) and those without (PS−) among individuals with 22q11.2 deletion syndrome (22q11DS) and its impact on functional outcome. The study found that individuals with 22q11DS and more prominent psychosis features exhibit a worsening of symptoms and functional decline, which is driven by neurocognitive decline, particularly related to executive functions and specifically working memory. Furthermore, differences in the trajectories of psychosis symptoms and neurocognitive performance were identified between individuals with more prominent psychosis features and those without such features, emphasizing the significance of evaluating and treating neurocognitive deficits in this population. [RQ1][RQ3][RQ4].

Furthermore, the final outcome of a study [101] focuses on neuroimaging of hippocampal subfields in schizophrenia and bipolar disorder. This study aimed to synthesize existing neuroimaging findings to better understand the structural changes in the hippocampal subfields associated with these disorders. The details of their findings, including the specific impacts on hippocampal structure and the implications for understanding the neurobiology of schizophrenia and bipolar disorder, are elaborated in the *Journal of Psychiatric Research*. [RQ1][RQ3][RQ4].

The final outcome of a study [102] focuses on the significant functional decoupling from the right ventral caudate to both occipital fusiform gyri. This study found that dopaminergic modulation induced significant functional decoupling in these brain regions, particularly in participants with low schizotypal personality scores. This suggests a potential link between dopamine-induced striato-occipital decoupling and schizotypal traits. [RQ1] [RQ3][RQ4].

## 5. Discussion

The paper emphasizes significant neuropsychological impairments across psychotic spectrum disorders, particularly focusing on memory, attention, and executive function. These cognitive deficits are highlighted as central to the pathology of disorders such as schizophrenia, bipolar disorder, and depression. Memory impairments are discussed in terms of both working memory and long-term memory deficits. Attentional deficits are noted to impact sustained and selective attention. Executive function impairments are discussed, including problems with planning, decision-making, and cognitive flexibility. These deficits are crucial for understanding the challenges faced by individuals with psychotic spectrum disorders and underscore the importance of targeted interventions.

The present study discusses the link between cognitive deficits and brain abnormalities in areas such as the prefrontal cortex, hippocampus, and thalamus. It emphasizes that structural and functional changes in these regions are closely related to the cognitive impairments observed in psychotic spectrum disorders. The prefrontal cortex is associated with executive functions and decision-making, the hippocampus with memory formation and retrieval, and the thalamus with sensory perception and attention regulation. The paper suggests that disruptions in these areas contribute significantly to the neuropsychological profile of these disorders, affecting memory, attention, and executive functioning.

Also, the present study emphasizes the diagnostic significance of neuropsychological tests for identifying cognitive deficiencies in individuals with psychotic disorders. It outlines how these tests are crucial in detecting early cognitive signs that may predict the onset and progression of conditions such as schizophrenia and bipolar disorder. Neuropsychological assessments are highlighted for their ability to provide detailed profiles of cognitive strengths and weaknesses, aiding clinicians in developing targeted treatment plans and interventions. More specifically, based on the systematic analysis of papers included in the study, the main findings could be summarized as follows:[RQ1] Cognitive impairments in memory, attention, and executive function are significantly more pronounced in individuals with psychotic spectrum disorders compared to the general population, and these impairments are correlated with structural and functional abnormalities in specific brain regions such as the prefrontal cortex, hippocampus, and thalamus.[RQ2] Neuropsychological tests are reliable diagnostic tools that can identify cognitive deficiencies early in the disease process of psychotic disorders and can predict the onset and progression of these conditions.[RQ3] Intervention strategies focused on cognitive rehabilitation can significantly improve cognitive functions in patients with psychotic spectrum disorders, highlighting the potential for new therapeutic approaches based on neuroplasticity and cognitive training.[RQ4] Genetic predispositions, combined with environmental stressors, significantly increase the risk of developing psychotic spectrum disorders, suggesting that early intervention and prevention strategies should target high-risk individuals with a familial history of these disorders.

It is worthwhile to note that the intricate relationship between the neurocognitive deficits observed in psychotic spectrum disorders and the underlying neural and molecular mechanisms of memory and learning presents a compelling area of study. Insights from recent studies [103,104] on targeting human glucocorticoid receptors in fear learning highlight the significant role of functional connectivity and stress hormones in modulating memory processes. This fundamental aspect could intersect meaningfully with cognitive impairments in psychosis. Furthermore, their work on the neural correlates and molecular mechanisms underlying memory and learning elucidates potential pathways that might be disrupted in psychotic disorders, offering a deeper understanding of the neurobiological foundations of these conditions. The integration of these perspectives with findings from the present study emphasizes the necessity of a multiscale approach in comprehending and addressing cognitive dysfunctions within psychotic spectrum disorders. This holistic view not only fosters a nuanced appreciation of the complexities involved but also paves the way for innovative interventions targeting specific molecular pathways and neural circuits implicated in both the neurodevelopmental trajectories of psychosis and the cognitive impairments that characterize these disorders.

The present findings aim to guide future research efforts towards understanding the neuropsychological underpinnings of psychotic spectrum disorders, improving diagnostic accuracy, and developing effective treatments. Additionally, the study strongly advocates for incorporating neuropsychological evaluations into the diagnostic process for psychotic spectrum disorders. It argues that such assessments can significantly enhance the precision of diagnoses and the customization of intervention strategies for cognitive impairments. Through detailed cognitive profiling, these evaluations enable a more nuanced understanding of each patient’s unique cognitive challenges, facilitating targeted therapeutic approaches and potentially improving treatment outcomes.

Addressing the limitations and suggesting avenues for future research based on this comprehensive systematic review, alongside discussing practical implications for clinical practice, is essential for the progression of the field. A primary limitation includes the potential underrepresentation of diverse populations in the studies, calling for future research to encompass a broader demographic to enhance the generalizability of findings. There is a significant need for longitudinal research to understand the neurodevelopmental trajectories of psychotic disorders better, as many included studies are cross-sectional. This could elucidate the progression of cognitive impairments and neurobiological changes, offering early intervention opportunities. The review indicates a need for deeper exploration of the causal mechanisms behind cognitive impairments and brain abnormalities. Employing advanced neuroimaging techniques could uncover specific neural circuits targeted by interventions. Moreover, integrating genetic and environmental factors is crucial, but their interactive effects on psychotic disorders remain unclear, suggesting a multidisciplinary approach for future studies. Acknowledging potential publication bias, as studies with negative or inconclusive results are less likely to be published, is critical. Future systematic reviews should include unpublished data and gray literature for a more unbiased field overview.

In clinical practice, enhancing diagnostic accuracy through neuropsychological assessments could lead to earlier and more precise categorization of psychotic spectrum disorders, facilitating tailored interventions. The importance of targeted therapies for cognitive impairments within psychotic disorders suggests that clinicians should consider cognitive rehabilitation and psychosocial interventions as part of comprehensive treatment plans. Given the observed heterogeneity in cognitive impairments and neurobiological patterns, intervention approaches should be personalized, incorporating not only pharmacological treatments but also noninvasive brain stimulation techniques and cognitive remediation therapies. Educating healthcare professionals and caregivers on the intricate relationship between dementia and psychosis and the complex neurobiological and cognitive impairments involved is crucial. Providing comprehensive education can enhance the care and support provided to individuals with psychotic spectrum disorders. By addressing these limitations and considering clinical practice’s practical implications, future research can advance our understanding and treatment of psychotic spectrum disorders, ultimately improving patient care and outcomes.

Future work arising from this systematic review should prioritize expanding the understanding of the neurobiological mechanisms involved in psychotic spectrum disorders. Particular focus is warranted on longitudinal studies that track the progression of cognitive deficits and the effectiveness of cognitive rehabilitation strategies over time. Research should also aim to develop and standardize neuropsychological assessments suited for use in low-resource settings. Further research is needed to develop validated, concise neuropsychological assessments for psychotic patients’ cognitive impairment in poor and middle-income countries. Future interventions could benefit from a precision medicine approach, tailoring therapies to the cognitive and neurobiological profiles of individual patients. Concomitantly, elucidating the complexities associated with the intersection of psychosis and dementia, childhood trauma, and the development of psychosis will be critical. Such research would considerably benefit from an integrative methodology that considers genetic, neuroimaging, and clinical data to unveil intricate interdependencies. It is important to note that the human mind is a complex network of cognitive processes, affective experiences, sensory interpretations, and behavioral responses carefully regulated by brain neurological mechanisms. 

## 6. Conclusions

This study underscores the significance of investigating the neuropsychological components of psychotic spectrum disorders. Identified cognitive impairments in attention, memory, and executive function realms signal crucial connections with structural and functional abnormalities present within key brain regions. Moreover, the correlation of dementia and psychosis amplifies the complex interplay between cognitive deterioration and psychotic symptoms and outlines the essentiality of targeted therapies. The multifactorial etiology of such disorders, encompassing genetic, environmental, and psychosocial factors, further necessitates precise diagnoses and personalized treatment modalities. Also, emphasized is the need for a deeper understanding of the neuropsychological processes in afflicted individuals which in turn enhances our grasp of the resultant impact on daily functionality. Greater insight into the interlinks between factors like childhood trauma, self-awareness, and cognitive functioning can bring to light the multidimensional intricacies of psychotic spectrum disorders and dementia, thus facilitating more effective therapeutic measures. The impact of antipsychotic medication on cognitive abilities in psychotic patients, especially those with Alzheimer’s, warrants further exploration to redefine pharmaceutical therapeutic approaches. In essence, a more comprehensive understanding of the neuropsychological aspects of psychotic spectrum disorders is central to formulating effective diagnostic and intervention strategies to enhance prognosis, diagnosis accuracy, and therapeutic outcomes.

To sum up, the synthesis of this comprehensive body of research underscores the complex neuropsychological landscape of psychotic spectrum disorders. The evidence demonstrates significant cognitive impairments, particularly in memory, attention, and executive functions, and reveals how these deficits are intertwined with neurobiological abnormalities and diagnostic challenges. The complex interplay between genetic risk factors, environmental triggers, and neurodevelopmental processes contributes to the onset and progression of these illnesses. The studies cited provide greater understanding of the neurocognitive and structural changes occurring in conditions such as schizophrenia, bipolar disorder, and related illnesses, particularly in the presence of additional factors like childhood trauma and dementia. This review draws attention to the potential of tailored neuropsychological assessments and interventions, and it highlights the importance of early identification and management of cognitive disruptions. The need for personalized treatment strategies that consider each individual’s unique neurobiological and psychological profile is evident. Future research should continue to advance the understanding of these disorders, refine diagnostic tools, and develop effective interventions that address not only the symptoms but also the cognitive and emotional challenges faced by individuals with psychotic spectrum disorders.

## Figures and Tables

**Figure 1 medicina-60-00645-f001:**
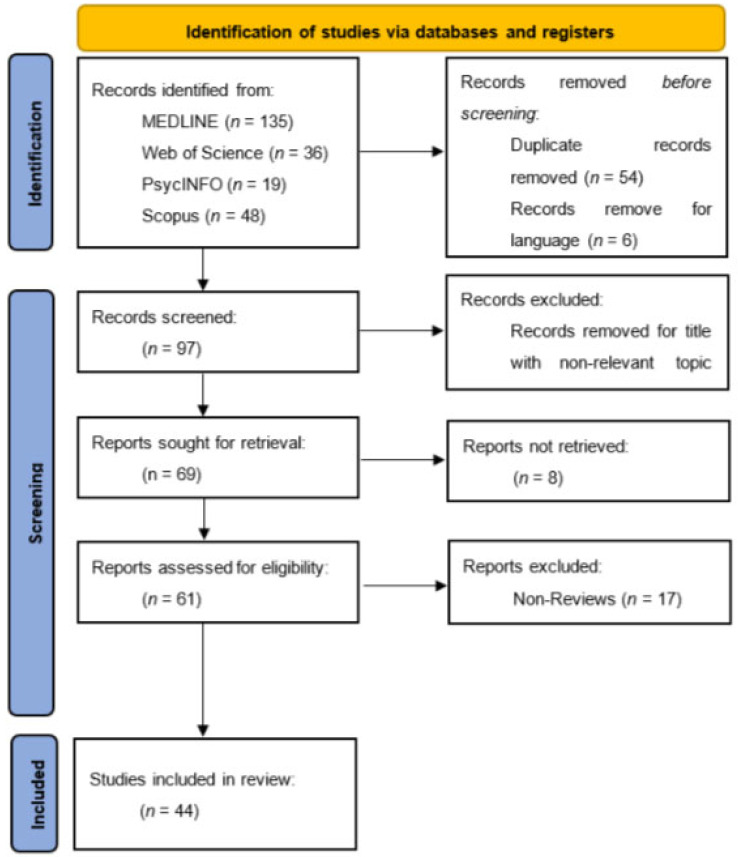
Flowchart of PRISMA Methodology.

**Table 1 medicina-60-00645-t001:** Studies of Systematic Analysis.

Authors	Sample	Outcomes Measured	Main Findings
Keshavan et al., 2020 [60]	Total: 446	The main or primary outcome measured in the study is the functional outcome in psychotic disorders, particularly related to cognitive impairments and mismatch negativity (MMN) as a biomarker of psychosis.	- There is considerable evidence for overlap in symptomatology and core cognitive impairments between different psychotic disorders and between these disorders, depression, and personality disorders. Cognitive impairments in bipolar disorder, especially in those with psychotic features, are associated with worse outcomes and a more chronic course. - Mismatch negativity (MMN) is a biomarker and a neurodegenerative marker of psychosis, with deterioration of MMN correlating with impairments in neurocognition and social cognition, as well as gray matter volume reduction in patients and clinical high-risk individuals who later transitioned to psychosis. MMN is a stronger indicator of functional outcome in early psychosis. - There is considerable biological heterogeneity across the psychosis syndromes, with the identification of three categories or “biotypes” across the psychosis dimension characterized by distinct alterations in cognition and electrophysiology, which appear to be orthogonal to the DSM-based clinical categorization.
Nelson et al., 2020 [61]	Total: 123 (Ultra-high risk for psychosis: 50—First-episode psychosis: 39—Healthy controls: 34)	Examination of Anomalous Self-Experience (EASE) scores. The primary outcomes of this study offer valuable insights into the initial stages of psychosis, underscoring the significance of incorporating both neurobiological and phenomenological dimensions in both research and clinical settings.	- Source monitoring explained a significant portion of the variance in EASE scores, indicating its potential relevance to minimal self-disturbance in schizophrenia spectrum disorders. - Aberrant salience measures were more strongly associated with general psychopathology measures, particularly positive psychotic symptoms, than with EASE scores. - The neurophenomenological model of minimal self-disturbance in schizophrenia spectrum disorders may need to be expanded to include other relevant constructs such as temporal processing, intermodal/multisensory integration, and hierarchical predictive processing.
Lewandowski et al., 2020 [62]	Total: 228	Neuroprogressive trajectories of neurocognition, structural brain measures, and network connectivity over the first eight years of illness and their predictive utility on clinical and functional outcomes.	- The main findings of the study emphasize the importance of characterizing neuroprogression in the early course of psychosis and its potential impact on the development of individualized treatments.
Culbreth et al., 2018 [63]	Total: 74	The main or primary outcome measured in the study is the association between late positive potential (LPP) amplitude and symptom severity in individuals diagnosed with affective psychosis and individuals with schizophrenia, including concurrent and prospective associations with symptoms.	- The late positive potential (LPP) did not show mean-level differences between individuals with schizophrenia spectrum disorders and those with primary affective psychosis. - In the primary affective psychosis group, reduced LPP amplitude was associated with greater depressive, negative, and psychotic symptom severity, both concurrently and at follow-up. - The results suggest that the neural correlates of emotion dysfunction may differ across psychotic disorders, with a possibility that schizophrenia is characterized by a decoupling of symptom severity and emotional processing.
Sugranyes et al., 2020 [64]	Total: 128 (SzO: 33—BpO: 46—Controls: 49)	Longitudinal changes in measures of brain structure, including cortical thickness, surface area, and gray matter volume, in children and adolescents at familial high risk (FHR) for bipolar disorder or schizophrenia who experienced psychotic spectrum symptoms over time.	- The emergence of psychotic spectrum symptoms in FHR was associated with smaller cross-sectional surface area and progressive cortical thinning. - Cortical thinning associated with illness could begin progressively during childhood and adolescence, possibly exhibiting early areas of overlap between SzO and BpO in the occipital cortex. - Both FHR individuals who developed psychotic spectrum symptoms and those who did not showed less time-related decrease in total surface area, indicating different trajectories of change in surface area over time between familial risk for schizophrenia versus bipolar disorder.
Bora, 2018 [65]	Total: 2047 (BD-I: 1211—BD-II: 836) Total: 1761 (PBD: 1017—NPBD: 744)	Cognitive impairment in different subgroups of bipolar disorder, including BD-I and BD-II, as well as subgroups based on history of psychosis (PBD and NPBD).	- Both history of psychosis and BD-I diagnosis were associated with modestly more pronounced global cognitive impairment.BD-I underperformed compared to BD-II in specific cognitive domains such as verbal memory, processing speed, EF-speed, and EF-accuracy. - PBD was associated with significantly impaired cognition compared to NPBD in various cognitive domains.The neurocognitive differences between clinical subtypes of BD are subtle and not distinctive, and most of the cognitive heterogeneity in BD cannot be explained by proposed subtypes of BD.
Wang et al., 2018 [66]	Total: 616 (PBD patients and 902 healthy subjects)	GMV differences between PBD patients and HC, specifically involving the prefronto-temporal and cingulate cortices, precentral gyrus, and insula.	- Psychotic bipolar disorder (PBD) patients exhibited smaller gray matter volumes (GMVs) in specific cortical regions compared to healthy controls, both qualitatively and quantitatively. - The higher proportions of female patients and patients taking psychotropic medication in PBD and PBD type I were associated with smaller GMV in specific cortical regions. - The study suggests that psychosis in BD might be associated with specific cortical GMV deficits, and gender and psychotropic medication might have effects on the regional GMVs in PBD patients.
McCleery and Nuechterlein, 2019 [67]	Total: 105	The main or primary outcome measured in the study is not explicitly stated. However, based on the content of the paper, the main or primary outcome measured seems to be the prevalence, profile, and magnitude of cognitive impairment in psychotic disorders, as well as the longitudinal stability of cognitive impairment.	- Cognitive impairment is a significant determinant of community functioning in individuals with psychotic disorders. - Most individuals with psychotic illness experience cognitive impairment. - Cognitive performance is a robust predictor of community functioning in people with psychotic disorders.
Vargas et al., 2018 [68]	Total: 3315	Association between childhood trauma and overall neurocognitive function in individuals with psychotic disorders. Relationship between childhood trauma and working memory in individuals with psychotic disorders.	- Significant association between overall cognition and childhood trauma in individuals with psychotic disorders. - Modest, negative relationship between childhood trauma and working memory in individuals with psychotic disorders. - Stronger association between childhood trauma and neurocognition in healthy controls compared to patients with a psychotic disorder.
Widmayer et al., 2019 [69]	Total: 334 patients and 113 controls (236 patients and 92 HC subjects)	Through the comparison of brain imaging data across different groups, the study sought to unveil the specific neural correlates of aggression in psychosis.	- Nonsystematic functional correlates of aggression in schizophrenia. - Few studies conducted with varied paradigms and overlapping samples. - No research on persons with affective psychoses.
Smucny et al., 2018 [70]	Total: 193 (SZ: 65—Schizophreniform: 2—Schizoaffective: 13—BD type I with psychotic features: 27—Healthy controls: 86)	Using advanced neurocognitive assessments and longitudinal study designs, researchers can delve deeper into the nature of cognitive impairments and their effects on individuals across different diagnostic categories.	- Deficits in cognitive control were present and stable over the early course of psychotic illness in both schizophrenia and bipolar disorder. - The study aimed to determine if previously identified deficits in cognitive control remained stable over a one-year period, and the results supported the hypothesis of stable cognitive control deficits in both schizophrenia and bipolar disorder. - The findings support the use of the AX-CPT paradigm to examine endophenotypic biomarkers of cognitive control in psychosis.
Hallford and Sharma, 2019 [71]	Total: 4221 (Schizophrenia spectrum: 3300—Major depression: 921)	Self-reported anticipatory pleasure in individuals with psychiatric disorders compared to control groups.	- Anticipatory pleasure is impaired in schizophrenia spectrum and major depression. Deficits in anticipatory pleasure are manifest in these disorders, and significantly more so in major depression. These findings indicate a possible therapeutic target to link cognitive, affective, and behavioral factors that precipitate and maintain disorder.
Dwyer et al., 2020 [72]	Total: 1223 (Discovery sample: 765—Validation sample: 458)	Subtype-specific illness courses including psychosis symptoms, depression symptoms, global functioning, and quality of life; polygenic scores for schizophrenia, bipolar disorder, major depression disorder, and educational achievement.	- The study identified five subgroups of psychosis with distinct illness courses and differences in educational attainment polygenic scores.
O’Neill et al., 2018 [73]	Total: 526 (combined HC groups), 420 (combined patient groups)	Functional connectivity (FC) of the default mode (DMN), salience (SN), and central executive networks (CENs) in patients with first-episode psychosis (FEP) compared to healthy controls.	- The study provides substantial evidence of widespread resting-state functional connectivity (FC) abnormalities of the default mode network (DMN), salience network (SN), and central executive network (CEN) in early psychosis, particularly implicating DMN and SN disconnectivity as a core deficit underlying the psychopathology of psychosis. - The DMN primarily displayed decreased connectivity between regions within the DMN, but also with regions in the SN and CEN. - The SN also displayed reduced connectivity with regions in the DMN and CEN, whilst demonstrating additional hyperconnectivity particularly with regions involved in visual and auditory processing.
Connors et al., 2018 [74]	Total: 445	The main or primary outcome measured in the study includes neuropsychiatric symptoms, dementia severity, cognition, function, caregiver burden, medication use, and mortality.	- Delusions and hallucinations independently and in combination are associated with poor clinical outcomes. The presence of both symptoms was associated with worse outcomes than only one of these symptoms. Antipsychotic medication use predicted mortality.
Dugré et al., 2020 [75]	Total: 2555	Functional brain deficits in individuals with conduct problems (CPs) and their adult form, adult antisocial behaviors, across distinct neurocognitive domains, including acute threat response, social cognition, cognitive control, and punishment and reward processing.	- Acute threat response: decreased activations in several brain regions; social cognition: altered activations in multiple brain regions; cognitive control: reduced activation in specific brain regions.
Vaquerizo-Serrano et al., 2021 [76]	Total: 16,474 (CHR-P: 875)	The primary outcome measured in the study is the presence of ASD in CHR-P individuals.	- 11.6% of CHR-P individuals have an ASD diagnosis.
Jonas et al., 2022 [77]	Total: 428	Preadmission cognitive scores extracted from school and medical records and postonset cognitive scores based on neuropsychological testing at 6-month, 24-month, 20-year, and 25-year follow-ups.	- The study observed three phases of cognitive change: normative, declining, and deteriorating. - Individuals with schizophrenia began to experience cognitive decline 14 years before psychosis onset at a significantly faster rate than those with other psychotic disorders. - Cognitive trajectories in schizophrenia were consistent with both a neurodevelopmental and neurodegenerative pattern, resulting in a loss of 16 IQ points over the period of observation, suggesting potential windows for primary and secondary prevention.
D’Antonio et al., 2019 [32]	Total: 40 (AD + P: 20—AD-P: 20—HC: 20)	Impairment in specific cognitive domains predicting the onset of psychosis in AD patients, gray matter alterations, their location, and the rate of atrophy associated with psychosis of AD.	- The presence of psychosis in AD is associated with specific cognitive deficits and gray matter alterations in specific brain regions. - The results provide new insights into the neuropsychological and neuroanatomical correlates of psychosis in AD.
Demro et al., 2022 [78]	Total: 332 (Schizophrenia: 105—Schizoaffective: 17—Bipolar I disorder with psychotic features: 41—First-degree biological relatives: 103—Controls: 66—Completed both studies: 42—Bipolar I disorder without psychosis: 15—Relatives of individuals with bipolar I disorder without psychosis: 7)	Advanced brain age measured using the Brain-Age Regression Analysis and Computation Utility Software (BARACUS, v1.1.2) prediction model, compared between individuals with a primary psychotic disorder and people with bipolar I disorder with a history of psychotic symptoms, as well as their biological first-degree relatives. The study also examined the association between advanced brain age and cognitive functioning, general functioning, and clinical diagnostic boundaries.	- The main findings of the study are that individuals with psychotic disorders demonstrated a larger brain-age gap compared to their biological relatives and healthy controls, indicating advanced brain age in schizophrenia and bipolar disorder. - Additionally, the study found no evidence of accelerated brain aging in psychotic psychopathology, suggesting early neurodevelopmental neural abnormalities. The findings support the presence of an abnormal neurodevelopmental process in psychotic disorders.
Voineskos et al., 2020 [79]	Total: 88	Primary outcome: cortical thickness in gray matter. Secondary outcome: microstructural integrity of white matter.	- The study found that exposure to olanzapine compared with placebo was associated with significant decreases in cortical thickness in the left and right hemispheres. - There was also a significant treatment group by time interaction in cortical thickness, and post hoc analyses showed that those who relapsed receiving placebo experienced decreases in cortical thickness compared with those who sustained remission.
Waszczuk et al., 2021 [80]	Total: 881	The presence of subtle abnormalities in white matter tracts connecting the frontal and temporal lobes, especially the SLF, ILF, and IFOF.	- Most studies indicate the presence of subtle abnormalities in white matter, particularly in the frontal and temporal lobes and their connections, such as the superior longitudinal fasciculus (SLF), inferior longitudinal fasciculus (ILF), and inferior fronto-occipital fasciculus (IFOF). - The comparison of DTI indices between schizophrenia patients and UHR individuals suggests the presence of subtle WM abnormalities prior to the onset of full-blown psychosis but reports on their severity and location differ. - Methodological factors, such as differences in age, sex, clinical presentation, or the drugs and psychoactive substances used by respondents, may contribute to the differences in the reviewed studies.
Bloomfield et al., 2021 [81]	Total: 24,793 (Clinical: 1639—Nonclinical: 23,154)	The potential roles of psychological processes in the associations between developmental trauma and specific psychotic experiences (i.e., hallucinations, delusions, and paranoia) in adulthood.	- Mediating roles of dissociation, emotional dysregulation, and PTSD symptoms between developmental trauma and hallucinations. - Evidence of a mediating role of negative schemata between developmental trauma and delusions as well as paranoia. - Distinct psychological pathways from developmental trauma to psychotic phenomena in adulthood.
Viher et al., 2021 [82]	Total: 83	Association between neurological soft signs (NSSs) and white matter alterations in adults with schizophrenia.	- The main findings indicate a positive association between neurological soft signs (NSSs) and diffusion measures in important motor pathways, reflecting the nature of NSSs at the interface of basic and higher-order motor control. This association suggests that structural and functional brain alterations may collectively explain the heterogeneous trajectory of NSSs in psychosis. - Additionally, the study found an association between NSSs in patients with schizophrenia and increased diffusivity in the corticospinal tract, corpus callosum, and superior longitudinal fascicle. These findings highlight the potential role of white matter alterations in motor pathways in contributing to NSSs in schizophrenia.
Ludwig et al., 2019 [83]	Total: 2498	Effectiveness of emotion regulation strategies in patients with psychotic disorders, specifically the association between maladaptive strategies and positive symptoms.	- Emotion regulation is markedly impaired in patients with psychotic disorders, with rumination and self-blaming identified as potential treatment targets.
Wu and Xiao, 2023 [84]	Total: 655	The primary outcome measured in the study is detecting brain abnormalities in diverse psychiatric illnesses with neuroimaging versus conventional methods.	- The study included 12 randomized controlled clinical trials with a substantial number of psychiatric patients, and it strongly recommends the use of neuroimaging techniques for the detection of psychiatric disorders.
Howes et al., 2018 [85]	Total: 38 (Schizophrenia: 16—Bipolar affective disorder: 22)	Striatal dopamine synthesis capacity (Kicer), correlation of Kicer with positive psychotic symptom severity.	- Elevated dopamine synthesis capacity is associated with psychosis across diagnostic boundaries and linked to the severity of psychotic symptoms. Striatal dopamine receptor density and structural gray matter volumes in various brain regions show heterogeneity in psychotic disorders, with frontal cortical regions demonstrating reduced heterogeneity.
Gama Marques and Ouakinin, 2019 [86]	Total: 192 (SCZ: 44—SAF: 44—Bipolar controls: 44—Follow-up patients: 60—SCZ: 30—SAF: 30)	Assessment of unconjugated bilirubin (UCB) as a biomarker for schizophrenia (SCZ) and schizoaffective (SAF) spectrum disorder during relapse and partial remission.	- There is a statistically significant difference in unconjugated bilirubin (UCB) levels between schizophrenia (SCZ) and schizoaffective (SAF) spectrum disorder patients during psychotic relapse, as well as between these patients and bipolar controls. - The research suggests potential for UCB as a biological marker for SCZ and SAF spectrum disorders during relapse and partial remission.
Anteraper et al., 2021 [87]	Total: 237 (CHR-: 144— CHR+: 23—HC: 93)	Resting-state functional connectivity (RsFc) differences in the dentate nuclei (DN) that may precede the onset of psychosis in individuals at risk of developing schizophrenia.	- Abnormalities in functional connectivity between the dentate nuclei (DN) and cerebral cortical areas may precede the onset of psychosis. - The study is the first to report abnormalities in functional connectivity of the DN within the cerebellum in individuals at risk for schizophrenia. - The results provide further support for a wide range of functional networks implicated in the pathophysiology of schizophrenia, including mechanisms of disease that precede conversion to psychosis in individuals at risk.
Koike et al., 2021 [88]	Total: 50 (Female: 23)	The relationship between neurocognitive deficits and improvements in UHR individuals and their association with symptom severity outcomes, as well as the paths from brain structural and functional characteristics to neurocognitive function and symptom severity outcomes.	- The main findings of the study include the associations between neurocognitive deficits and brain activity, as well as cortical structure, in UHR individuals. - Additionally, the study highlighted the negative associations between verbal fluency deficits and negative symptoms, as well as processing speed deficits and excitement symptoms. - Lastly, the study revealed significant paths from specific cortical surface areas to verbal fluency deficits and short-term negative symptoms.
Ruiz et al., 2020 [89]	Total: 2205	Effort failure rate and moderators of effort test. A comprehensive analysis of neuropsychological effort test performance in individuals with psychotic disorders, providing valuable insights into their cognitive profiles.	- Psychotic disorders are associated with a generalized neurocognitive deficit, effort test failure rate is associated with global neuropsychological impairment, and effort tests have questionable validity in this population.
Kim et al., 2021 [90]	Total: 64 (FEP: 35—Healthy Controls: 29)	The correlation between ToM strange story scores and the FA values of the left cingulum and left SLF in patients with FEP.	- The study demonstrated the white matter connectivity underlying the mentalizing network and its relation to ToM ability in patients with FEP. - Patients with FEP exhibited impaired ToM abilities, as indicated by the results of the false belief task and strange story task. - Positive associations were found between the integrity of specific white matter regions (left ROIs) and ToM deficits in FEP patients.
Torrent et al., 2018 [91]	Total: 192 (Nonaffective psychoses: 142—Affective psychoses: 50)	Functioning at follow-up, assessed by a regression model composed of PANSS total score and verbal fluency assessed by the FAS (COWAT).	- Reduced performances in executive functions at baseline combined with symptom severity were predictors of poor functional outcome in patients with first psychotic episodes. - Clinical and neurocognitive differences observed at baseline decreased over the two-year follow-up period.
Stein et al., 2022 [92]	Total: 1071	Association of FTD dimensions with GMV and FA, establishment of a transdiagnostic factor model of FTD, and linking psychopathological factors to brain structural measures across disorders.	- The study revealed a three-factor model of formal thought disorder (FTD) across disorders, comprising disorganization, incoherence, and emptiness. - Disorganization was associated with parts of the temporo-occipital language junction, while emptiness showed a negative correlation with specific brain regions including the hippocampus and thalamus. - Disorganization and incoherence were differentially associated with white matter structures, indicating common neurobiological structures involved in FTD across affective and psychotic disorders.
Muetzel et al., 2018 [93]	Total: 845	Association between psychiatric symptoms (externalizing and internalizing) at baseline and the changes in subcortical gray matter volume and global fractional anisotropy over time.	- Higher ratings for externalizing and internalizing symptoms at baseline predicted smaller increases in both subcortical gray matter volume and global fractional anisotropy over time. - Children presenting with behavioral problems at an early age show differential subcortical and white matter development. - The study demonstrates a link between psychiatric problems along a continuum and a differential pattern of brain changes over time.
Lepage et al., 2020 [94]	Total: 80	Change in cortical thickness and volume of the hippocampus as a function of the duration of unremitted positive symptoms.	- Cortical thinning in specific brain regions was significantly associated with the duration of unremitted psychotic symptoms during the first interscan interval, but not during the second interscan interval or with hippocampal volumes. - The study suggests that psychotic symptoms may lead to cortical reorganization early in the disease course of psychosis.
Papanastasiou et al., 2018 [95]	Total: 1434 (High PLEs: 149—Low PLEs: 149)	Brain activation during a monetary incentive delay reward task in healthy adolescents at ages 14 and 19 years old.	- Alterations in prefrontal and striatal function during reward processing may be involved in the development of psychosis, and the nonclinical sample in the study may reflect a combination of aberrant salience leading to abnormal experiences and a compensatory cognitive control mechanism necessary to contextualize them.
De Picker et al., 2020 [96]	Total: 101 (Patients: 49—Healthy control subjects: 52)	Identification of state and trait markers in the peripheral immune system and two immune-associated neuroendocrine pathways (IDO and GTP-CH1 pathways) in a longitudinal sample of psychosis patients.	- Patients with acute psychosis had significantly higher plasma concentrations of proinflammatory markers such as CRP, CCL2, IL1RA, IL6, IL8, and TNFα and lower concentrations of neuroendocrine pathway markers such as KA and KA/Kyn. These markers normalized after treatment. - The levels of nitrite, another immune marker, increased sharply after the initiation of antipsychotic medication. - Positive symptoms during the acute episode correlated with proinflammatory markers, while negative symptoms correlated inversely with IDO pathway markers. - Normalization of KA and 3-HK levels corresponded to a larger improvement of negative symptoms, and decreasing KA levels were associated with relative improvement in Symbol–Digit Substitution Task (SDST) scores.
Kuipers et al., 2018 [97]	Total: 8580 (Follow-up subsample: 2406)	Persecutory ideation, hallucinations, affective symptoms, effects of cannabis and problematic alcohol use.	- Worry has a central role in the links between symptoms, with direct effects on insomnia, depressed mood, generalized anxiety, and recent cannabis use. - The reciprocal influence of worry and paranoia implies that treating either symptom is likely to ameliorate the other. - The paper’s novel statistical analysis using dynamic Bayesian networks enabled the re-examination of relationships between interacting affective and psychotic variables over time.
Sánchez-Morla et al., 2018 [98]	Total: 139 (Euthymic bipolar patients: 99—Healthy controls: 40)	Change in neurocognitive composite index (NCI) over a 5-year follow-up period, specifically in relation to the number of manic and hypomanic episodes experienced by bipolar patients, as well as its association with working memory and visual memory.	- The progression of cognitive decline is not a general rule in BD. - BD patients with a greater number of manic or hypomanic episodes may constitute a subgroup characterized by the progression of neurocognitive impairment. - Prevention of manic and hypomanic episodes could have a positive impact on the trajectory of cognitive function.
Chendo et al., 2022 [99]	Total: 2919 for psychosis and 3161 for any form of hallucination	Frequency of psychosis and any form of hallucination in PD patients.	- Around 20% of Parkinson’s disease patients experience psychosis or hallucinations. - The risk of developing hallucinations is likely influenced by disease duration, Hoehn and Yahr stage, and cognitive status.
Gur et al., 2023 [100]	Total: 157 (PS+: 98—PS-: 59)	Differences in the trajectories of psychosis symptoms and neurocognitive performance between the PS+ and PS- groups and the impact on functional outcome.	- Individuals with 22q11DS and more prominent psychosis features show worsening of symptoms and functional decline driven by neurocognitive decline, particularly related to executive functions and specifically working memory. - Differences in the trajectories of psychosis symptoms and neurocognitive performance were identified between individuals with more prominent psychosis features and those without such features. - Neurocognitive decline was found to drive the clinical change, highlighting the importance of evaluating and treating neurocognitive deficits in this population.
Haukvik et al., 2018 [101]	Total: 2393 (Schizophrenia patients: 909—Bipolar disorder patients: 625—Healthy controls: 1089)	Hippocampal subfield volumes or shape in schizophrenia and bipolar disorder.	- The development of automated hippocampal subfield segmentation methods enables detailed characterization of hippocampus regions and identification of subregion involvement in schizophrenia and bipolar disorder. Schizophrenia and bipolar disorder have widespread hippocampal subfield volume reductions. - Patients with schizophrenia had smaller left CA2/3 and right presubiculum compared to those with bipolar disorder. Both schizophrenia and bipolar disorder affect hippocampal subfield morphology, but schizophrenia has greater effects.
Rössler et al., 2018 [102]	Total: 54 (L-DOPA: 33—Placebo: 32)	The main or primary outcome measured in the study is the significant functional decoupling from the right ventral caudate to both occipital fusiform gyri.	- Dopaminergic modulation induced significant functional decoupling from the right ventral caudate to both occipital fusiform gyri, particularly in participants with low schizotypal personality scores. This suggests a potential link between dopamine-induced striato-occipital decoupling and schizotypal traits.

**Table 2 medicina-60-00645-t002:** Neuropsychological Assessment for Dementia and Psychosis.

Neuropsychological Test	Cognitive Domain Assessed	Relevance to Dementia and Psychosis
Wechsler Memory Scale (WMS)	Memory and learning	Essential for assessing both episodic and working memory deficits common in dementia and psychosis.
Trail Making Test (TMT) Parts A and B	Attention, processing speed, executive function	Useful for evaluating cognitive flexibility and processing speed, which are often impaired in both conditions.
Wisconsin Card Sorting Test (WCST)	Executive function, problem-solving	Helps in assessing abstract thinking and the ability to shift cognitive strategies in response to changing environmental contingencies, challenges seen in both dementia and psychosis.
Rey–Osterrieth Complex Figure Test (ROCFT)	Visuoconstructive abilities, memory	Assesses the ability to organize and remember complex visual information, reflecting on visuoperceptual and executive deficits.
Verbal Fluency Tests (Semantic and Phonemic)	Language, executive function	Evaluates language function and executive processes related to generating strategies for retrieval, often affected by both dementia and psychosis.
Digit Span Test (Forward and Backward)	Attention, working memory	Measures attentional capacity and working memory, crucial for understanding the extent of cognitive decline.
Stroop Color and Word Test	Attention, processing speed, executive function	Assesses cognitive flexibility and susceptibility to interference, a common issue in cognitive impairments associated with dementia and psychosis.

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
