# Peer review of "Integrating Clinical Neuropsychology and Psychotic Spectrum Disorders: A Systematic Analysis of Cognitive Dynamics, Interventions, and Underlying Mechanisms"

_medicina, 2024, doi:10.3390/medicina60040645_

Round 1
Reviewer 1 Report
Comments and Suggestions for Authors
In this systematic review, Gkintoni et al. illustrate the neuropsychological impairments across psychotic spectrum disorders and explore the potential cognitive impairments, which show a close connection to brain abnormalities. The authors also highlight that neuropsychological tests may play a critical role in detecting cognitive deficiencies. It would be great to further discuss which neuropsychological tests may be beneficial (adding a table could be helpful).
Overall, I would like to consider this study an important contribution of interest to a large readership. Therefore, this manuscript should be accepted in its present form.
Author Response
Dear Reviewer,
Thank you for your valuable comments and the time devoted to ameliorate our manuscript. We hightlight the role of neuropsychological testing as indicated and provided a table with common used tests and batteries for psychotic spectrum and dementia.
On behalf of the authorship
Kind regards
Evgenia Gkintoni
Reviewer 2 Report
Comments and Suggestions for Authors
This paper titled ‘Aspects of Clinical Neuropsychology Related to the Psychotic 2 Spectrum. A Comprehensive Systematic Review’ provides a comprehensive examination of neuropsychological impairments within psychotic spectrum disorders, emphasizing the significant impact of cognitive deficits on the daily functioning and overall quality of life of affected individuals. It highlights the importance of understanding specific cognitive impairments, such as memory, attention, executive functioning, and language abilities, to develop effective treatments and improve prognosis for those with psychotic spectrum disorders.
A key focus is on the neuropsychological functioning of dementia patients with psychosis, exploring the intersection of cognitive impairments and psychotic symptoms. The paper discusses the prevalence of traumatic experiences and their potential impact on cognitive functioning and the development of psychotic disorders. It also addresses the limited understanding and knowledge surrounding psychosis, particularly schizophrenia, and the implications for treatment adherence and outcomes.
The methodology section outlines a systematic literature review conducted according to PRISMA guidelines, focusing on clinical neuropsychology related to the psychotic spectrum. The review included articles published between 2018 and 2023, resulting in the inclusion of 44 papers for analysis. Findings from the reviewed literature suggest a significant association between childhood trauma and neurocognitive function in individuals with psychotic disorders, with implications for working memory and overall cognition. The paper also notes the lack of systematic studies on the functional correlates of aggression in psychosis, highlighting a gap in the literature. Finally, the discussion emphasizes the central role of memory, attention, and executive function impairments in the pathology of psychotic spectrum disorders, including schizophrenia, bipolar disorder, and depression. It points to structural and functional brain abnormalities in key regions, such as the prefrontal cortex, hippocampus, and thalamus, as closely related to these cognitive deficits.
In conclusion, the paper underscores the critical importance of investigating neuropsychological components in psychotic spectrum disorders. It calls for future research to expand understanding of neurobiological mechanisms, develop standardized neuropsychological assessments for low-resource settings, and explore precision medicine approaches for tailored therapies. The study highlights the complex interplay between cognitive deterioration and psychotic symptoms, advocating for precise diagnoses and personalized treatment modalities to address the multifactorial etiology of these disorders.
In general, I think the idea of this review article is really interesting and the authors’ fascinating observations on this timely topic may be of interest to the readers of Medicina. However, some comments, as well as some crucial evidence that should be included to support the author’s argumentation, needed to be addressed to improve the quality of the manuscript, its adequacy, and its readability prior to the publication in the present form. My overall judgment is to publish this paper after the authors have carefully considered my suggestions below, in particular reshaping parts of the ‘Introduction’ and ‘Methods’ sections by adding more evidence.
Please consider the following comments:
- A graphical abstract that will visually summarize the main findings of the manuscript is highly recommended.
- I would ask the authors to ensure that the abstract clearly outlines the significance of studying the intersection between dementia and psychosis and highlights the unique contributions of the research. Also, I think they should include specific findings related to cognitive impairments, neuropsychological assessments, and the impact of psychosis on individuals with dementia.
- Introduction: In my opinion, in this section authors should clearly delineate the existing gap in literature concerning the intricate relationship between dementia and psychosis, emphasizing the need for further exploration in this area. For example, I would suggest better elaborating on the critical importance of comprehending this intersection for healthcare professionals and caregivers, underlining how a deeper understanding can lead to more effective care strategies and improved patient outcomes, as well as delving into the neural substrates implicated in the interaction between dementia and psychosis. Exploring the underlying neural mechanisms could offer valuable insights into how cognitive impairments and psychotic symptoms manifest at a neurobiological level, potentially paving the way for targeted interventions that address these complex conditions more effectively.
- Literature Review: While utilizing databases like Scopus, PsycINFO, PubMed, and WoS is commendable, I would ask the authors to provide a detailed description of the search strategy, including keywords used and search filters applied, to enhance transparency and reproducibility. Also, as this is a systematic review, authors should acknowledge potential publication bias and discuss how this might impact the review's findings.
- I would suggest a better discussion on any unexpected or contradictory results and their implications for the study. Authors should address limitations explicitly and suggest avenues for future research to build upon the current findings, but also discuss practical implications of the reviewed literature for clinical practice, such as implications for diagnostic accuracy, treatment strategies, and intervention approaches in individuals with psychotic spectrum disorders.
- According to the Journals’ guidelines, Journal references must cite the full title of the paper, page range or article number, and digital object identifier (DOI) where available. Please, correct the actual Reference list accordingly.
- Please correct Table 1 format: according to the Journals’ guidelines, to facilitate the copy-editing of larger tables, smaller fonts may be used, but no less than 8 pt. in size. Authors should use the Table option of Microsoft Word to create tables.
I hope that, after careful revisions, the manuscript can meet the journal’s high standards for publication. I declare no conflict of interest regarding this manuscript.
Best regards,
Reviewer
References:
1. https://doi.org/10.3390/ijms25020864
2. https://doi.org/10.3390/ijms25052724
Comments on the Quality of English Language
Minor English editing is required.
Author Response
Dear Reviewer,
Thank you for the valuable comments and the time devoted to ameliorate our manuscript. We try addressing all of your comments and the revisions are highlighted in green color. We emphasized the correlation with dementia and psychotic spectrum. The introduction is reformatted significantly as indicated. The literature review has been reorganized in its large part as well as the discussion as indicated. Furthermore, the references suggested, have been added enriching the content of the manuscript. Thank you a lot!
Thank you once again for the constructive collaboration!
On behalf of the authorship
Evgenia Gkintoni
Reviewer 3 Report
Comments and Suggestions for Authors
"Aspects of Clinical Neuropsychology Related to the Psychotic Spectrum. A Comprehensive Systematic Review" is an attempt in reviewing the role of neuropsychology in psyhcosis. A clear structure is missing, e.g.g the definition what neuropsychology is, is provided in line 412ff.
There are lots pf repetitions, sudden jumps, e.g. the paragraph is on dementia when suddenly neuroleptic drugs are discussed, and sections that read as if a BA student was asked to summarise a paper, e.g. line 380 to line 391
for a systematic review no literature (PRISMA) or other guidelines are used. Keywords and databases used are not reported.
After 10 pages (that could be cut to 2 pages) the authors provide 4 (practially more than 4) research questions.
If the authors follow that strucutre they should first review how cognitive impairments manifest in psychotic distorders (behavioural perspective). Then the neurobiological perspective (PFC, HPC, thalamus etc). Regarding diagnostic importance and predicting onset and progression, the question addresses sensitivity and specificity of the tests. Then - and less relevant to the title of the paper - the authors ask about efficacy of intervention strategies and rehabilitation. This is not the focus of the paper given the title. The last RQ on genetic and environmental factors as risk factors is not focal.
If the authors restrict themselves to their RQ1 and RQ2, streamline the paper and provide a systematic review (http://www.prisma-statement.org/) there might be value in this review for practitioners. The meothd section does state the use of PRISMA, but there is no information about the Boolean terms, the "reports excluded" states non-reviews, i.e. the systematic review is a review of reviews
The result section is not clear. It lacks structure. Please order by your RQs, i.e. which reviews (of the 44) answers RQ1 and so on and what is their finding. What do x papers / reviews say about cognitive impairment in psychosis? It is very hard to follow wjhat you are writing as you describe details of papers (i.e. mismatch negativity, late positive potential) but not the overarching concepts of memory, attention, executive functions etc
Given table 1, it is clear that you are not using review articles but original articles. Then only having 44 articles in your search is surprisingly low.
Your aim to include FHR, CHR-P or UHR or even relatives is too ambitious. The literature is hige. There are many studies investigating cognitive / behavioural and biological markers for at risk for psychosis people.
Valuable would be a focus on dementia in psychosis - restrict your review to that. What is known about it? Cut out childhood trauma (could well be a contributing factor, but the focus of your review should be on "if the person has a diagnosis of psychosis, are they at risk for dementia" - what do we know about that?
and maybe add, "if a person has dementia, is this associated with psychotic symptoms+" suggesting some biological mechanism / link between cognitive impairments and psychosis
minor: reference 85 is all in capital letters
To
Comments on the Quality of English Language
Your writing style is not very good.
E.g. "indicated by the research conducted by researchers (line 369)"
there are many similar examples, and you can easily save 50% of your words in the introduction and over 30% throughout the manuscript.
Author Response
Dear Reviewer,
Thank you for the valuable comments and the time devoted to ameliorate our manuscript. We addressed all the comments as indicated. All the revisions suggested are highlighted in green color.
Kind regards
Evgenia Gkintoni
Round 2
Reviewer 3 Report
Comments and Suggestions for Authors
Thank you for improving the method part by now reporting the Boolean search terms.
Unfortunately, you still have over 10 pages before the method section. PLesae cut this to maximally 3 pages, you repeat yourself a lot. The reader does not need that much repetition. It suffice to explain that deficits, discovered with neuropsychological testing, suggest that reduced cognitive abilities contribute to psychosis. Here we will show that patients with other diseases known for cognitive decline (e.g., dementia) can develop / have psychotic-like experiences. ...
If you streamline your review, from over 30 pages (the table is neat, thank you) to 20ish pages, it will receive more readers. But as it is, it is a long passage before one gets to the actual review.
In table 2 you also have a few referneces with no entrance in column "outcome measured", or provide only a very cryptic formulation of it. It is advisable if you classify outcome measured into the cognitive abilities of memory, executive function, attention etc as this aligns better with your paper (your table 1)
Comments on the Quality of English LanguageMy comment on English is again style, not grammar or typos.
Author Response
Dear Reviewer,
Thank you for your valuable comments.
Regarding the manuscript has been shortened as indicated in the sections introduction, results and discussion and many repetitions have eliminated.
Also in the table with results have been added some elements in the column outcome measures.
Thank you once again for the constructive collaboration.
Kind regards
Evgenia Gkintoni